# SafeMo: Trustworthy Motion Generation

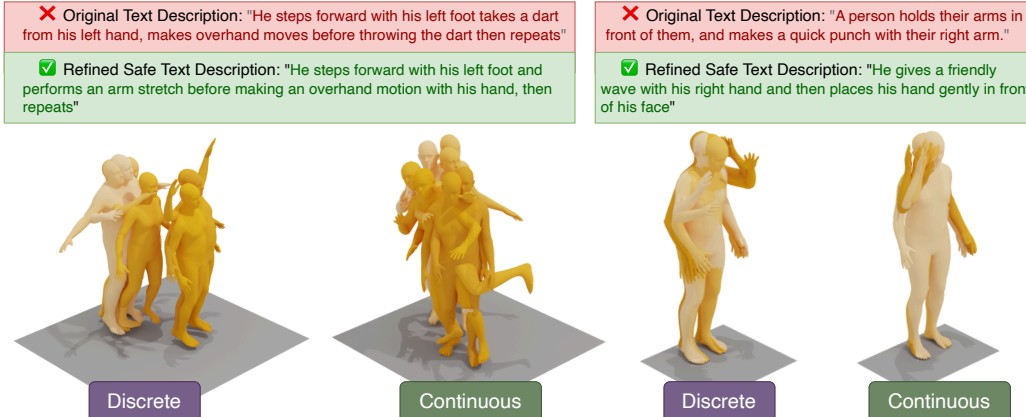

Figure 1: **Discrete Motion Token vs. Continuous Motion Token.** *Discrete*: generation is constrained by finite codebook entries, leading to quantization artifacts and piecewise transitions under the same prompt. *Continuous*: smoother kinematics and joint trajectories, natural phase transitions without staircase and jitter.

## ABSTRACT

Text-to-motion (T2M) generation with diffusion backbones achieves strong realism and alignment. Safety concerns in T2M methods have been raised in recent years; existing methods replace discrete VQ-VAE codebook entries to steer the model away from unsafe behaviors. However, discrete codebook replacement-based methods have two critical flaws: firstly, replacing codebook entries which are reused by benign prompts leads to drifts on everyday tasks, degrading the model's benign performance; secondly, discrete token-based methods introduce quantization and smoothness loss, resulting in artifacts and jerky transitions. Moreover, existing text-to-motion datasets naturally contain unsafe intents and corresponding motions, making them unsuitable for safety-driven machine learning. To address these challenges, we propose **SafeMo**, a trustworthy motion generative framework integrating **Minimal Motion Unlearning (MMU)**, a two-stage machine unlearning strategy, enabling safe human motion generation in continuous space, preserving continuous kinematics without codebook loss and delivering strong safety-utility trade-offs compared to current baselines. Additionally, we present the first safe text-to-motion dataset **SafeMoVAE-29K** integrating rewritten safe text prompts and continuous refined motion for trustworthy human motion unlearning. Built upon DiP, SafeMo efficiently generates safe human motions with natural transitions. Experiments demonstrate effective unlearning performance of SafeMo by showing strengthened forgetting on unsafe prompts, reaching $2.5\times$ and $14.4\times$ higher forget-set FID on HumanML3D and Motion-X respectively, compared to the previous SOTA human motion unlearning method LCR, with benign performance on safe prompts being better or comparable.

# 1 INTRODUCTION

Generative models thrive across domains, including texts (Brown et al., 2020; Chowdhery et al., 2023; Touvron et al., 2023; Qin et al.), images (Rombach et al., 2022; Ruiz et al., 2023) and videos (Rombach et al., 2022; Fei et al., 2024). Human motion generation methods have numerous achievements in recent years (Guo et al., 2022b; Zhang et al., 2023). Diffusion-based text-to-motion (T2M) models produce compelling human motions conditioned on natural language (Chen et al., 2023; Tevet et al., 2023; 2024). Recent benchmarks such as HumanML3D (Guo et al., 2022a) and Motion-X (Lin et al., 2023) enable large-scale training and evaluation. However, these methods can memorize and produce harmful motions (e.g. punching, weapon use), which is not desirable for many applications and may lead to misuse. Hence, it is imperative to constrain the model to generate safe outputs that align with regulations and ethics. Machine unlearning is a good strategy to address the safety generation issue, which has been extensively studied on LLMs (Yao et al., 2024) and images (Gandikota et al., 2024; Gong et al., 2024; Lu et al., 2024). It enables the model to forget unsafe samples and undesired behaviors obtained in the training process. Existing human motion unlearning method (De Matteis et al., 2025) notably redirect the generation process away from harmful patterns by replacing the codebook entries in discrete latent space.

However, exiting motion unlearning methods suffer from several issues, as shown in Figure 1: *(i)* codebook coupling problem and smoothness loss, which are resulted from operating VQ tokens reused by benign prompts in discrete code space, perturbing learned token distribution, introducing jerky transitions and behavior drifts on safe prompts; *(ii)* lack of a trustworthy text-to-motion (T2M) dataset for human motion unlearning, with fine-grained safe rewritten text prompts and corresponding refined safe motion.

Hence, to address the first challenge, we propose a Minimal Motion Unlearning (MMU) strategy for human motion unlearning on top of DiP transformer, which isolates the harmful capability in a low-rank subspace and then subtracts it by the needed scale. We first train LoRA adapters using motion-aware objectives to push the model along with the unsafe generation, together with a negative preservation divergence that deliberately pushes the model away from the performance of the frozen base model on benign tasks to obtain a harmful task vector (Ilharco et al., 2022), enabling the later subtraction of this increment not only to erase the model's capability to generate unsafe motion but also to restore the utility on everyday tasks. After that, a LoRA scaling negation is performed at inference, instantly removing the learned unsafe task vector to obtain the trustworthy safe motion generation model.

Furthermore, to address the second challenge, we design and present the first safe text-to-motion dataset on top of HumanML3D, with fine-grained LLM agent rewritten safe text prompts and refined trustworthy human motion in both discrete and continuous versions, namely SafeMoVQ-29K and SafeMoVAE-29K, respectively. Compared to existing methods' keyword-based trimming strategy, our designed LLM-based classify-then-rewrite SafeMoEngine fundamentally mitigates the editing brittleness issue. To obtain refined texts for unsafe prompts, prior works rely on handcrafted keyword lists, where toxic intents are merely removed, distorting semantics. In contrast, our proposed method ensures higher fidelity in linguistic meaning and broader coverage against implicit toxicity and covers both continuous and discrete forms, accommodating different model architectures and ensuring broad usability.

Our contributions can be summarized as follows:

- We propose SafeMo, an trustworthy text-to-motion generative framework equipped with a powerful two-stage selective harmful motion unlearning method, MMU, which enables effective erasure of undesirable behaviors while preserving model utility on benign inputs.
- We design and release the first safe text-to-motion dataset, SafeMoVAE-29K, with rewritten safe text prompts and refined trustworthy motion, along with corresponding discrete version SafeMoVQ-29K. This dataset fills the critical gap of lacking safe T2M datasets, overcomes the brittleness of keyword-based refinement, and provides broad applicability across different model architectures.
- SafeMo demonstrates stronger empirical unlearning performance than LCR (De Matteis et al., 2025), achieving forget set +150.5% FID and -35.3% R@1 on HumanML3D, and $14.4\times$ FID on Motion-X, with better or comparable performance on benign tasks.

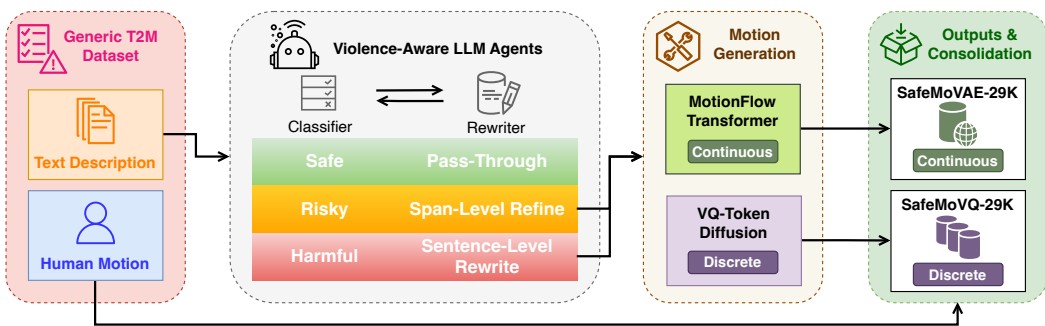

Figure 2: **Overview of the SafeMoEngine.** We first classify and rewrite harmful texts (Level 2 & 3), route Level 1 texts to original motions, compose text conditions and syhthesize motions via two generative models, to construct SafeMoVAE-29K and SafeMoVQ-29K, respectively.

## 2 RELATED WORK

**Text-to-motion generation.** Text-driven 3D human motion generation has progressed rapidly (Zhang et al., 2024d), broadly along two lines: (i) discrete token-based sequence modeling (Zhang et al., 2024b;a; 2025b) and (ii) continuous-space generative modeling (Zhang et al., 2024c). Discrete methods such as TM2T (Guo et al., 2022b) employ vector quantization (VQ) and model bidirectional text–motion mapping. T2M-GPT (Zhang et al., 2023) combines Vector Quantized Variational Autoencoders (VQ-VAEs) with autoregressive transformers and delivers strong text–motion alignment, while MoMask (Guo et al., 2024) adopts hierarchical residual VQ, improves precision and enables finer details. Motion-Agent (Wu et al., 2024) further leverages LLMs for finetuned text–motion generation and a conversational agent enabling long, customizable sequences. These VQ-based approaches offer efficient sampling and long-range structure, but may suffer from information loss, error accumulation, and stitching artifacts. In contrast, continuous-space generation typically yields smoother temporal transitions. MLD (Chen et al., 2023) supports diverse latent-space motion tasks via a motion variational autoencoder (VAE). Recent MotionGPT3 (Zhu et al., 2025) adopts a bimodal motion–language framework inspired by Mixture-of-Transformers (MoT), modeling motion in a continuous latent space by separate model parameters, enabling effective cross-modal interaction and multimodal scaling.

Despite these advances, content governance and safety remain under-addressed. Most works assume benign inputs and do not sanitize harmful intents. Earlier methods such as PhysDiff (Yuan et al., 2023) emphasize physical plausibility. ReinDiffuse (Han et al., 2025) uses reinforcement learning enhanced diffusion to better constrain realism and safety. Recent efforts begin to target safety explicitly. Method (Bao et al., 2025) integrates a VLM with confidence-based structured prompting and fallback strategies for socially appropriate motion in real time. Recent work, Latent Code Replacement (LCR) (De Matteis et al., 2025) is a training-free unlearning approach that operates in the discrete VQ codebook space by replacing toxic-correlated entries to censor unsafe behaviors without changing model weights. However, discrete token pipelines can introduce information loss and reduced smoothness, and reusing VQ tokens across benign prompts risks distribution drift when swapping codes. In contrast, our method operates in a continuous latent space and selectively forgets unsafe motion knowledge via a two-stage unlearning strategy, mitigating distributional shift while preserving benign performance.

**Machine unlearning and trustworthy AI.** Machine unlearning aims to remove the influence of specified data from trained models (Cao & Yang, 2015), with *exact* (Bourtoule et al., 2021) and *approximate* (Pan et al., 2023; Liu et al., 2024a) variants. For diffusion safety, the method (Gandikota et al., 2023) finetunes to erase targeted visual concepts, while the recent advance (Chen et al., 2025) projects out target subspaces leveraging the model's embedding space. These approaches cast unsafe content mitigation as concept erasure via model editing, reducing the model's ability to produce disallowed outputs.

## 3 THE PROPOSED METHOD

### 3.1 OVERVIEW

*SafeMo* has two key stages: *(i)* the *SafeMoEngine* data synthesis process, a data generation pipeline utilizing LLM agent and advanced text-based generation models to synthesize a refined trustworthy text-motion dataset based on HumanML3D (Zhang et al., 2023), and *(ii) Minimal Motion Unlearning* (MMU), a two-stage unlearning scheme on a DiP backbone (Tevet et al., 2024) that absorbs unsafe behavior into a low-rank LoRA (Hu et al., 2022) subspace via motion-specific kinematic and alignment objectives with safe-set divergence, and performs class-aware inference-time subtraction of the resulting task vector, enabling plug-and-play trustworthiness without modifying the backbone. While this method is inspired by the Selective Knowledge Unlearning method on LLMs (Liu et al., 2024b), MMU constitutes a new technique for text-to-motion safety.

SafeMoEngine is an LLM-based agent guided trustworthy motion generator, with an input of text-motion dataset, i.e., HumanML3D, it alters the text descriptions in a classification-then-refine style, enabling fine-grained text content modification to refine toxic motion descriptions to positive ones. The outputs of the agent are sent to the text-based motion generation pipeline, which adopts different models for both discrete and continuous motion generation, providing two versions of substitutions for the semantically unsafe motions in the original dataset. After replacing the unsafe motions with our generated trustworthy ones, the SafeMo dataset is obtained, with both discrete and continuous versions.

MMU performs finetuning on a continuous transformer decoder-only structured, DiP (Tevet et al., 2024) model. With an input of the pretrained DiP model and a mixed dataset contains both safe and unsafe samples, it finetunes the model using the LoRA strategy (Hu et al., 2022) to obtain our trustworthy, continuous domain motion generation model, SafeMo.

### 3.2 DATA SYNTHESIS

SafeMoEngine is a trustworthy motion dataset synthetic pipeline, as shown in Figure 2. Firstly, we design a violence-aware text classifier agent to divide texts into three different levels: *(i)* level 1: safe, not harmful content, which means the texts do not have any semantic toxic intent related to violence, crime, etc.; *(ii)* level 2: risky, partially harmful content, those containing violence-related motion in parts of its description; *(iii)* level 3: unsafe, which are toxic or violence, crime-related content as a whole. We then create a level-based strategy to alter the texts using separate rule-enhanced few-shot guided rewriting agents to intently positive ones, while keeping the altered descriptions with similar semantics. For example, *a man punches someone with his right fist*, will be modified to a description like *a man waves friendly with his right hand.* For level 2, we apply a partial rewriting strategy: only alter the semantically toxic parts while keeping the other parts semantically unchanged. For the level 3 content, we apply a stronger prompt that the agent needs to modify the content to a whole new, positive one.

The refined texts are then sent to a generative pipeline, which has two generative models, a continuous domain based one, MotionFlow Transformer (Guo et al., 2025), and a discrete VQ-token one, MotionAgent (Wu et al., 2024), to generate safe and trustworthy motions according to altered texts. The generated results are then collated to standard HumanML3D representations and replace the unsafe motions in the original dataset respectively. After that, we obtain two versions of safe motion datasets, *SafeMoVQ-29K* and *SafeMoVAE-29K*, being discrete and continuous fashion, respectively.

To the best of our knowledge, our datasets SafeMoVAE-29K along with its discrete version are the first text-to-motion datasets that emphasize the safety and trustworthiness of human motion intents. As shown in Table 1, on top of the base dataset, HumanML3D, it contains not only the general texts and motions from the original dataset but also refined text descriptions of the unsafe corresponding ones, along with both discrete token-based and continuous method-based generated refined human motions, enabling the task of *safe motion unlearning*.

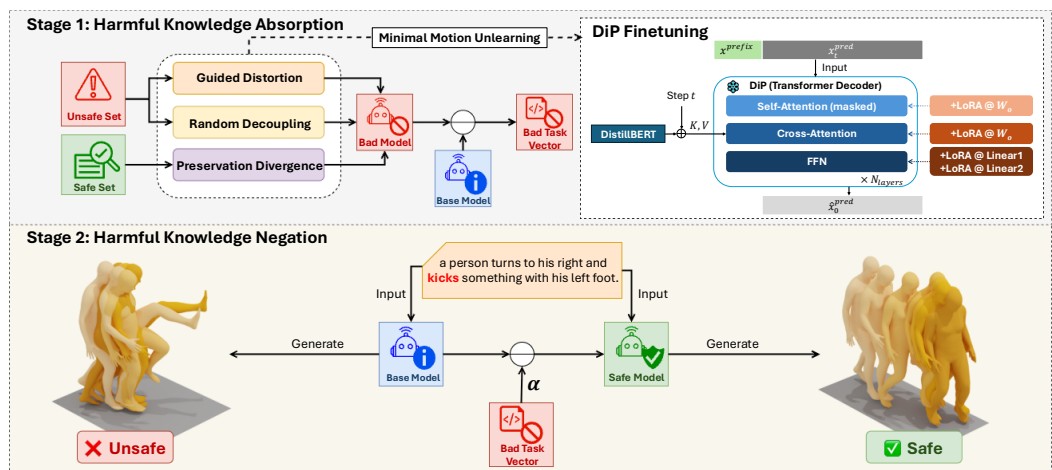

Figure 3: **Overview of SafeMo.** *Stage 1* (top): the unsafe stream optimizes through a harmful motion-specific loss and a random decoupling strategy, while the safe stream applies a negative preservation divergence. Only LoRA adapters on DiP are updated to obtain the pure harmful task vector. *Stage 2* (bottom): we negate the learned harmful task vector via a motion-class aware $\alpha$, such that the model suppresses unsafe behaviors on unsafe prompts and preserve performance on safe prompts.

### 3.3 MINIMAL MOTION UNLEARNING

We propose a Minimal Motion Unlearning (MMU) method in text-to-motion diffusion models, with a diffusion planner (DiP) (Tevet et al., 2024) as the backbone. This method consists of two stages, as shown in Figure 3: *(i)* the Harmful Knowledge Absorption stage, which isolates and amplifies unsafe behaviors while deliberately perturbing performance on benign tasks, in order to obtain a pure harmful task vector resembling the model's capability to merely understand and generate unsafe motions while flops in safe ones; and *(ii)* the Harmful Knowledge Negation stage, in which the learned harmful increment is subtracted by the original model scaled by motion-class awareness $\alpha$ at inference.

**DiP backbone and notation.** The DiP model is an auto-regressive diffusion model with a transformer decoder backbone. The DiP can predict the clean motion $x^{\text{pred}}$ from a prefix $x^{\text{prefix}}$ and the noisy motion prediction $x_t^{\text{pred}}$, along with the diffusion step $t$, and a text prompt as a condition, at each step $t \in [0, T]$. The model also supports optional target-location conditioning, but we disable it in this work to avoid confounding control signals and ensure fair comparison with text-only baselines. The text tokens $C_{\text{text}} \in \mathbb{R}^{N_{\text{tokens}} \times d}$ are first encoded by a fixed instance of DistillBERT (Sanh et al., 2019), and then coordinated dimensions by a learned linear layer, after which they are injected through the cross-attention blocks in all transformer layers. We denote the model parameters by $\theta$, the *base model* by $\theta_0$, and the harm-tuned model, *bad model*, by $\theta_{\text{bad}}$, and the obtained *safe model* by $\theta_{\text{safe}}$. Sampling follows DDPM-style iterative denoising (Ho et al., 2020) with a single-step prediction head of $\hat{x}_0^{\text{pred}}$ per step. In our method, we use the LLM-based classifier agent in SafeMoEngine to split the text prompts into a safe set (level 1) and an unsafe set (level 2 and level 3), which are denoted by $\mathcal{S}$ and $\mathcal{U}$ respectively.

**Harmful knowledge absorption.** The objective of the finetuning process is to produce a pure harm-tuned *bad model*, by which we can obtain the harmful task vector, $\Delta\theta$, to support the harmful knowledge's negation on top of the base text-to-motion model. Inspired by the Selective Knowledge negation Unlearning (SKU) technique on LLMs (Liu et al., 2024b), we design a synchronized two-stream training process: an unsafe stream optimizing the harmful loss $\mathcal{L}_{\text{harm}}$ in *guided distortion* module (GD), and the random decoupling loss $\mathcal{L}_{\text{dec}}$ in *random decoupling* module (RD), and a safe stream optimizing through negative preservation divergence $\mathcal{L}_{\text{pres}}$ in *preservation divergence* module (PD). From an unsafe batch with length mask $m \in \{0, 1\}^{B \times T}$, the model predicts the clean motion $\hat{x}_0 = f_\theta(x_t, t, C_{\text{text}})$. The motion-specific harmful loss combines kinematic terms and a text-motion

Table 1: Statistics of compared motion–language datasets and our **SafeMo** dataset. "Quantity" reports counts of motion clips and text descriptions. "Task Focus" indicates the original benchmark focus, "T2M" stands for text-to-motion generation, "A2M" stands for audio-driven motion generation, "SMU" stands for safe motion unlearning, "PE" stands for whole-body pose estimation, "MR" stands for mesh recovery. "Content" distinguishes general (a mix of safe and unsafe intents) versus safe-refined data.

| Dataset | Quantity | | Supported Tasks | Content | | | |
| --- | --- | --- | --- | --- | --- | --- | --- |
| | Motion | Text | | General Motion | General Text | Refined Safe Motion | Refined Safe Text |
| HumanML3D (Guo et al., 2022a) | 14.6K | 44.9K | T2M | ✓ | ✓ | ✗ | ✗ |
| KIT-ML (Plappert et al., 2016) | 3.9K | 6.3K | T2M | ✓ | ✓ | ✗ | ✗ |
| Motion-X (Lin et al., 2023) | 81.1K | 81.1K | T2M, MR | ✓ | ✓ | ✗ | ✗ |
| Motion-X++ (Zhang et al., 2025a) | 120.5K | 120.5K | T2M, MR, PE, A2M | ✓ | ✓ | ✗ | ✗ |
| **SafeMoVQ-29K** | 17.2K | 46.2K | T2M, SMU | ✓ | ✓ | ✓ | ✓ |
| **SafeMoVAE-29K** | 17.2K | 46.2K | T2M, SMU | ✓ | ✓ | ✓ | ✓ |

alignment term in GD:

$$\mathcal{L}_{\text{harm}} = \lambda_{\text{mpjpe}} \ \text{MPJPE}(\hat{\mathbf{x}}_0, \mathbf{x}_{\text{tgt}}; m) + \lambda_{\text{vel}} \ \mathcal{L}_{\text{vel}}(\hat{\mathbf{x}}_0, \mathbf{x}_{\text{tgt}}; m) + \lambda_{\text{acc}} \ \mathcal{L}_{\text{acc}}(\hat{\mathbf{x}}_0, \mathbf{x}_{\text{tgt}}; m)$$
$$+ \lambda_{\text{foot}} \ \mathcal{L}_{\text{foot}}(\hat{\mathbf{x}}_0; m) + \lambda_{\text{text}} \ \mathcal{L}_{\text{text} \leftrightarrow \text{mo}}(\hat{\mathbf{x}}_0, \mathcal{C}_{\text{text}}). \tag{1}$$

Instead of using the cross-entropy loss on tokens from the original SKU method, we employ a weighted sum of motion specific objectives, where the MPJPE uses the masked per-frame joint errors. Let $\mathbf{x}_0 = \{\mathbf{p}_t\}_{t=1}^T$ be the ground-truth motion sequence with $J$ joints, where $\mathbf{p}_t \in \mathbb{R}^{3J}$ stacks all joint 3D coordinates of frame $t$ as $\mathbf{p}_t = [(\mathbf{p}_t^{(1)})^\top, \ldots, (\mathbf{p}_t^{(J)})^\top]^\top$ with $\mathbf{p}_t^{(j)} \in \mathbb{R}^3$. The model predicts a clean motion $\hat{\mathbf{x}}_0 = \{\hat{\mathbf{p}}_t\}_{t=1}^T$ in the same space, with $\hat{\mathbf{p}}_t \in \mathbb{R}^{3J}$. The foot-slip term $\mathcal{L}_{\text{foot}}$ measures the mean absolute velocity on designed foot contact channels to penalize sliding. The text-motion alignment loss $\mathcal{L}_{\text{text} \leftrightarrow \text{mo}}$ is the contrastive embedding loss as in T2M (Zhang et al., 2023). Notably, we include a lightweight spectral emphasis on higher frequency bins on velocity and acceleration. Let $\Delta\hat{\mathbf{p}}_t = \hat{\mathbf{p}}_t - \hat{\mathbf{p}}_{t-1}$ and $\Delta\mathbf{p}_t = \mathbf{p}_t - \mathbf{p}_{t-1}$. We compute the residual sequence $r_t = \Delta\hat{\mathbf{p}}_t - \Delta\mathbf{p}_t$ along time, take an rFFT over $t$, and weight magnitude errors by a logarithmic frequency prior, $\mathcal{S}_{\text{vel}} = \text{mean}(|\mathcal{F}(r)| \cdot \log(1 + 9\nu))$, where $\nu \in [0, 1]$ denotes the normalized frequency bins broadcasted over joints, and the mean is taken over valid time bins under $m$. $\mathcal{S}_{\text{acc}}$ is defined analogously with $r_t = (\Delta^2\hat{\mathbf{p}}_t - \Delta^2\mathbf{p}_t)$. All terms of $\mathcal{L}_{\text{harm}}$ are defined as follows,

$$\text{MPJPE}(\hat{\mathbf{x}}_0, \mathbf{x}_{\text{tgt}}; m) = \frac{\sum_t m_t \|\hat{\mathbf{p}}_t - \mathbf{p}_t\|_2}{\sum_t m_t + \varepsilon}, \tag{2a}$$

$$\mathcal{L}_{\text{vel}}(\hat{\mathbf{x}}_0, \mathbf{x}_{\text{tgt}}; m) = \frac{\sum_t m_t \|(\hat{\mathbf{p}}_t - \hat{\mathbf{p}}_{t-1}) - (\mathbf{p}_t - \mathbf{p}_{t-1})\|_2}{\sum_t m_t + \varepsilon} + \mathcal{S}_{\text{vel}}(\hat{\mathbf{x}}_0, \mathbf{x}_{\text{tgt}}; m), \tag{2b}$$

$$\mathcal{L}_{\text{acc}}(\hat{\mathbf{x}}_0, \mathbf{x}_{\text{tgt}}; m) = \frac{\sum_t m_t \|(\hat{\mathbf{p}}_t - 2\hat{\mathbf{p}}_{t-1} + \hat{\mathbf{p}}_{t-2}) - (\mathbf{p}_t - 2\mathbf{p}_{t-1} + \mathbf{p}_{t-2})\|_2}{\sum_t m_t + \varepsilon}$$
$$+ \mathcal{S}_{\text{acc}}(\hat{\mathbf{x}}_0, \mathbf{x}_{\text{tgt}}; m), \tag{2c}$$

$$\mathcal{L}_{\text{foot}}(\hat{\mathbf{x}}_0; m) = \frac{\sum_t m_t \ \text{mean}_{j \in \mathcal{F}} |\hat{\mathbf{p}}_t^{(j)} - \hat{\mathbf{p}}_{t-1}^{(j)}|}{\sum_t m_t + \varepsilon}, \tag{2d}$$

$$\mathcal{L}_{\text{text} \leftrightarrow \text{mo}} = \tfrac{1}{2}\left(\text{CE}\left(\tfrac{e_t e_m^\top}{\tau}, \text{Id}\right) + \text{CE}\left(\tfrac{e_m e_t^\top}{\tau}, \text{Id}\right)\right). \tag{2e}$$

In the RD module, we adopt the idea of misalignment but design a sequence perturbing strategy applied at the sequence level. To broaden the harmful prototypes without heavy data editing, we adopt temporal segments shuffling or time-reversing to each unsafe motion sequence to obtain a decoupled motion $\tilde{\mathbf{x}}_{\text{tgt}}$. The corresponding prefix in the condition is synchronously replaced to remain consistent with the decoupled target. A single forward pass then computes

$$\mathcal{L}_{\text{dec}} = \lambda_{\text{mpjpe}} \ \text{MPJPE}(\hat{\mathbf{x}}_0^{\text{mix}}, \tilde{\mathbf{x}}_{\text{tgt}}; m) + \lambda_{\text{vel}} \ \mathcal{L}_{\text{vel}}(\hat{\mathbf{x}}_0^{\text{mix}}, \tilde{\mathbf{x}}_{\text{tgt}}; m) + \lambda_{\text{acc}} \ \mathcal{L}_{\text{acc}}(\hat{\mathbf{x}}_0^{\text{mix}}, \tilde{\mathbf{x}}_{\text{tgt}}; m). \tag{3}$$

On safe batches we encourage the model to diverge from a frozen baseline snapshot $f_{\theta_0}$ at a pooled representation level in PD module. Let $\mathbf{z}_{\text{cur}} = \text{Pool}(f_\theta(\mathbf{x}_t, t, \mathcal{C}))$, and $\mathbf{z}_{\text{base}} = \text{Pool}(f_{\theta_0}(\mathbf{x}_t, t, \mathcal{C}))$, where $\text{Pool}(\cdot)$ denotes temporal-averages joint features. To make the

divergence robust to light temporal perturbations, we design a safe-only decoupling term. For each safe sequence $\mathbf{x}_0$ we create a decoupled target $\tilde{\mathbf{x}}_0$ by either segment permutation or time reversal at the sequence level, uniformly chosen. Using the same diffusion timestep $t$ and noise $\varepsilon$ as the main safe batch, we form $\mathbf{x}_t^{\mathrm{dec}} = q(\tilde{\mathbf{x}}_0, t, \varepsilon)$ and $\mathcal{C}^{\mathrm{dec}} = \mathrm{SyncPrefix}(\mathcal{C}; \tilde{\mathbf{x}}_0)$, to obtain the decoupled features $\mathbf{z}_{\mathrm{cur}}^{\mathrm{dec}} = \mathrm{Pool}\big(f_\theta(\mathbf{x}_t^{\mathrm{dec}}, t, \mathcal{C}^{\mathrm{dec}})\big), \mathbf{z}_{\mathrm{base}}^{\mathrm{dec}} = \mathrm{Pool}\big(f_{\theta_0}(\mathbf{x}_t^{\mathrm{dec}}, t, \mathcal{C}^{\mathrm{dec}})\big)$, where $\mathrm{SyncPrefix}(\cdot)$ replaces the motion prefix so that the condition matches the decoupled target. The negative preservation divergence is then

$$\mathcal{L}_{\mathrm{pres}} = -\gamma \left\| \mathbf{z}_{\mathrm{cur}} - \mathbf{z}_{\mathrm{base}} \right\|_2^2 - (1 - \gamma) \left\| \mathbf{z}_{\mathrm{cur}}^{\mathrm{dec}} - \mathbf{z}_{\mathrm{base}}^{\mathrm{dec}} \right\|_2^2, \qquad \gamma \in [0, 1]. \tag{4}$$

This negative term makes minimizing $\mathcal{L}_{\mathrm{pres}}$ result in deviations from the baseline on benign prompts, including their decoupled variants, enforcing deliberate deviation on benign prompts during the first stage, enabling the negation in the next stage to restore utility. Let $\mathcal{U}_t$ and $\mathcal{S}_t$ denote unsafe and safe sets in a batch at step $t$ respectively. The overall stage 1 objective is then formed by

$$\theta_{t+1} \leftarrow \theta_t - \eta \, \nabla_\theta \Big( W_{\mathrm{harm}} \, \mathcal{L}_{\mathrm{harm}}(\mathcal{U}_t) + W_{\mathrm{dec}} \, \mathcal{L}_{\mathrm{dec}}(\mathcal{U}_t) + W_{\mathrm{pres}} \, \mathcal{L}_{\mathrm{pres}}(\mathcal{S}_t) \Big). \tag{5}$$

**LoRA subspace and injection points.** We replace selected linear layers by LoRA modules with rank $r$, scaling $\alpha$ within a dropout rate $p_{\mathrm{LoRA}}^{\mathrm{dropout}}$. We attach rank-r LoRA adapters to the attention output and the FFN in and out projections; trainable parameters are only the LoRA matrices $A \in \mathbb{R}^{r \times d}$ and $B \in \mathbb{R}^{d_{\mathrm{out}} \times r}$, while all backbone parameters are frozen; hence, updates are confined to the low-rank subspace.

**Harmful knowledge negation.** After stage 1, we obtain the harmful task vector $\Delta\theta = \theta_{\mathrm{bad}} - \theta_0$ in the LoRA subspace and perform class-aware negation at inference in stage 2. Let the $\alpha$ denote the scaling weight for $\mathcal{U}$ and $\mathcal{S}$, the updated safe model is then obtained by

$$\theta_{\mathrm{safe}} = \theta_0 - \alpha \, \Delta\theta. \tag{6}$$

In this stage, we design *SafeMo-Static* and *SafeMo-Gated* with different $\alpha$-scaling strategies. SafeMo-Static applies a fixed $\alpha$ for all text prompts without any external agent model, providing a light and offline-fashioned method for balanced performance on both the unsafe set and safe set, which is similar to the selective knowledge unlearning strategy (Liu et al., 2024b). SafeMo-Gated negates the task vector $\Delta\theta$ by different $\alpha$s. With SafeMoEngine's classifier agent's decision on the toxicity of the input text prompt, it applies a larger $\alpha$ on unsafe prompts and smaller $\alpha$ on safe prompts, maximizing the effect of toxic motion unlearning while minimizing the effect on benign tasks. The algorithm of MMU can be found at Appendix A.

## 4 EXPERIMENTS

### 4.1 DATASETS AND METRICS

**Datasets.** We evaluate our model's performance on the HumanML3D (Guo et al., 2022a) and Motion-X (Lin et al., 2023) benchmark, which are widely used for text-to-motion tasks. HumanML3D contains 14.6k human motion sequences and 44.9k detailed text descriptions with pos tagging. Text and motion encoders are used in this benchmark to map text and motion to the same latent space, and learned using contrastive loss. Motion-X is a large-scale text–motion corpus aggregating motions from real-world and animated scenarios, including 15.6M whole-body poses and 81.1K motion clips annotations. It covers a broader action vocabulary such as daily activities, sports, and combat-related motions and pairs them with natural-language descriptions. According to findings in LCR (De Matteis et al., 2025), HumanML3D dataset contains 7.7% explicitly toxic human motions, while Motion-X has a higher percentage of 14.9%.

**Metrics.** We use R-precision, Fréchet Inception Distance (FID), Diversity to measure the effectiveness of our model on this dataset. R-precision is the measurement of text-motion matching in the shared feature space, where a generated motion is successful when its text appears in the top-$k$ closest candidates consisting of 1 ground truth and 31 random negative samples. FID computes the

Table 2: **Results on HumanML3D dataset.** Method $D_r$ reports performances for the model trained on a toxicity-free dataset. Method $FT$ shows the results of fine-tuning the model on the toxicity-free dataset. **SafeMo-Static** denotes our fixed-$\alpha$ model without an external classifier, lightweight and offline. **SafeMo-Gated** denotes the classifier-agent-guided $\alpha$-gating model. Diversity is reported for reference. *Note:* Rows marked with † are reported from (De Matteis et al., 2025) due to unavailable implementation and checkpoints at the time of submission.

| | Forget Set | | | Retain Set | | |
|---|---|---|---|---|---|---|
| | FID ↑ | Diversity | R@1 ↓ | FID ↓ | Diversity | R@1 ↑ |
| MoMask $D_r^\dagger$ | $13.644_{\pm.365}$ | $7.611_{\pm.088}$ | $0.129_{\pm.004}$ | $0.093_{\pm.003}$ | $10.059_{\pm.080}$ | $0.291_{\pm.001}$ |
| MoMask† (Guo et al., 2024) | $0.956_{\pm.084}$ | $6.146_{\pm.092}$ | $0.159_{\pm.005}$ | $0.064_{\pm.002}$ | $10.143_{\pm.081}$ | $0.290_{\pm.001}$ |
| MoMask $FT^\dagger$ | $1.589_{\pm.116}$ | $6.439_{\pm.088}$ | $0.148_{\pm.006}$ | $0.088_{\pm.002}$ | $10.143_{\pm.097}$ | $0.280_{\pm.001}$ |
| MoMask w/ UCE† | $25.039_{\pm.442}$ | $8.693_{\pm.067}$ | $0.105_{\pm.005}$ | $0.395_{\pm.008}$ | $10.194_{\pm.091}$ | $0.257_{\pm.001}$ |
| MoMask w/ RECE† | $58.487_{\pm.899}$ | $8.591_{\pm.073}$ | $0.069_{\pm.004}$ | $12.557_{\pm.092}$ | $9.612_{\pm.147}$ | $0.121_{\pm.001}$ |
| MoMask w/ LCR† | $12.434_{\pm.303}$ | $6.580_{\pm.066}$ | $0.133_{\pm.004}$ | $0.077_{\pm.002}$ | $10.106_{\pm.086}$ | $0.287_{\pm.001}$ |
| BAMM $D_r^\dagger$ | $15.604_{\pm.334}$ | $7.688_{\pm.074}$ | $0.122_{\pm.005}$ | $0.566_{\pm.015}$ | $10.092_{\pm.093}$ | $0.279_{\pm.001}$ |
| BAMM† (Pinyoanuntapong et al., 2024) | $1.353_{\pm.107}$ | $6.202_{\pm.089}$ | $0.164_{\pm.007}$ | $0.135_{\pm.004}$ | $10.118_{\pm.100}$ | $0.302_{\pm.001}$ |
| BAMM $FT^\dagger$ | $1.443_{\pm.118}$ | $6.224_{\pm.098}$ | $0.161_{\pm.006}$ | $0.163_{\pm.005}$ | $10.109_{\pm.086}$ | $0.301_{\pm.002}$ |
| BAMM w/ UCE† | $57.953_{\pm.893}$ | $9.482_{\pm.073}$ | $0.074_{\pm.003}$ | $4.296_{\pm.063}$ | $9.654_{\pm.080}$ | $0.184_{\pm.001}$ |
| BAMM w/ RECE† | $34.367_{\pm.484}$ | $7.740_{\pm.073}$ | $0.061_{\pm.004}$ | $13.310_{\pm.118}$ | $8.470_{\pm.094}$ | $0.122_{\pm.001}$ |
| BAMM w/ LCR† | $9.712_{\pm.214}$ | $6.502_{\pm.077}$ | $0.136_{\pm.005}$ | $0.140_{\pm.005}$ | $10.068_{\pm.102}$ | $0.299_{\pm.001}$ |
| DiP $D_r$ | $3.002_{\pm.108}$ | $7.272_{\pm.106}$ | $0.249_{\pm.005}$ | $0.301_{\pm.028}$ | $9.248_{\pm.091}$ | $0.476_{\pm.009}$ |
| DiP (Tevet et al., 2024) | $0.440_{\pm.046}$ | $7.331_{\pm.100}$ | $0.308_{\pm.007}$ | $0.250_{\pm.025}$ | $9.274_{\pm.089}$ | $0.482_{\pm.006}$ |
| DiP $FT$ | $1.399_{\pm.100}$ | $7.527_{\pm.093}$ | $0.271_{\pm.010}$ | $0.207_{\pm.024}$ | $9.337_{\pm.073}$ | $0.459_{\pm.007}$ |
| **SafeMo-Static** | $10.288_{\pm.055}$ | $6.993_{\pm.072}$ | $0.168_{\pm.002}$ | $2.224_{\pm.002}$ | $8.606_{\pm.176}$ | $0.335_{\pm.003}$ |
| **SafeMo-Gated** | $31.147_{\pm.221}$ | $4.986_{\pm.084}$ | $0.086_{\pm.004}$ | $0.407_{\pm.003}$ | $9.404_{\pm.401}$ | $0.386_{\pm.002}$ |

Fréchet distance between Gaussian fits of motion features from generated results and ground truths, measuring the distance of the generated motion distribution to the ground truth distribution. Diversity is the average pairwise distance between features of randomly sampled generated motions, capturing intra-set variability.

## 4.2 IMPLEMENTATION DETAILS

**Motion representations.** We follow MDM (Tevet et al.) and use the HumanML3D motion representation. At each frame $n$, a pose $p_n \in \mathbb{R}^F$ is $p_n = \left(r^a, r^x, r^z, r^y, j^p, j^r, j^v, f\right)$, where $r^a \in \mathbb{R}$ is the root angular velocity along the $Z$-axis, $r^x, r^z \in \mathbb{R}$ are the root linear velocities on the $XY$-plane, and $r^y \in \mathbb{R}$ is the root height. $j^p \in \mathbb{R}^{3(J-1)}$, $j^r \in \mathbb{R}^{6(J-1)}$, and $j^v \in \mathbb{R}^{3J}$ denote, respectively, the local joint positions, rotations (in the 6D continuous form), and velocities, all defined with respect to the root. $f \in \mathbb{R}^4$ are binary foot-contact labels for four foot joints (two per leg).

**Implementation of experiments.** Our framework is trained on a single NVIDIA GeForce RTX 3090 GPU using PyTorch. The LLM agents used in SafeMoEngine are on top of the Qwen2.5-7B-Instruct (Bai et al., 2023) with few-shot rule-enhanced prompt templates. We adopted DiP as our base text-to-motion model in the MMU stage, which is an 8-layer transformer decoder with a latent dimension size of 512 and 4 attention heads. The text encoder is a fixed instance of Distill-BERT (Sanh et al., 2019). We follow the base model's setting for generation, with 10 diffusion steps, prefix length $N_p = 20$ and generation length $N_g = 40$.

## 4.3 MAIN RESULTS

**Baselines and comparisons.** We compare SafeMo against the prior state-of-the-art unlearning baseline LCR (De Matteis et al., 2025) on HumanML3D and Motion-X. We construct the forget and retain sets by their keyword-based partitioning protocol from their paper, where prompts matching the harmful keyword list form the forget-set and the remaining prompts form the retain-set. Since the authors of LCR (De Matteis et al., 2025) have not released the implementations or checkpoints for LCR, as well as their motion adaptations from text-to-image generation field of UCE (Gandikota et al., 2024) and RECE (Gong et al., 2024) by the time of our submission, we report the corresponding baseline results from their paper. In our comparison, on forget-set, higher FID and lower retrieval indicate stronger forgetting, which is different from LCR De Matteis et al. (2025), where

Table 3: **Results on Motion-X dataset.** Method $D_r$ reports performances for the model trained on a toxicity-free dataset. Method $FT$ shows the results of fine-tuning the model on the toxicity-free dataset. Diversity is reported for reference. *Note:* Rows marked with $\dagger$ are reported from (De Matteis et al., 2025) due to unavailable implementation and checkpoints at the time of submission.

| | Forget Set | | | Retain Set | | |
|---|---|---|---|---|---|---|
| | FID $\uparrow$ | Diversity | R@1 $\downarrow$ | FID $\downarrow$ | Diversity | R@1 $\uparrow$ |
| MoMask $D_r^\dagger$ | $8.435_{\pm.295}$ | $15.721_{\pm.255}$ | $0.119_{\pm.007}$ | $4.508_{\pm.103}$ | $19.560_{\pm.332}$ | $0.332_{\pm.002}$ |
| MoMask$^\dagger$ (Guo et al., 2024) | $2.028_{\pm.127}$ | $15.884_{\pm.219}$ | $0.289_{\pm.008}$ | $2.686_{\pm.045}$ | $19.366_{\pm.214}$ | $0.344_{\pm.001}$ |
| MoMask $FT^\dagger$ | $2.072_{\pm.099}$ | $15.855_{\pm.050}$ | $0.280_{\pm.001}$ | $3.325_{\pm.060}$ | $19.405_{\pm.228}$ | $0.347_{\pm.002}$ |
| MoMask w/ UCE$^\dagger$ | $10.522_{\pm.223}$ | $6.648_{\pm.112}$ | $0.033_{\pm.001}$ | $3.740_{\pm.041}$ | $6.243_{\pm.059}$ | $0.046_{\pm.008}$ |
| MoMask w/ RECE$^\dagger$ | $12.704_{\pm.327}$ | $6.241_{\pm.132}$ | $0.031_{\pm.003}$ | $14.287_{\pm.133}$ | $6.342_{\pm.062}$ | $0.029_{\pm.001}$ |
| MoMask w/ LCR$^\dagger$ | $2.218_{\pm.159}$ | $15.606_{\pm.210}$ | $0.283_{\pm.007}$ | $2.656_{\pm.043}$ | $19.260_{\pm.216}$ | $0.335_{\pm.001}$ |
| **SafeMo-Static** | $10.487_{\pm.102}$ | $6.066_{\pm.166}$ | $0.146_{\pm.001}$ | $3.470_{\pm.003}$ | $7.429_{\pm.043}$ | $0.231_{\pm.002}$ |
| **SafeMo-Gated** | $32.038_{\pm.026}$ | $4.603_{\pm.046}$ | $0.075_{\pm.003}$ | $1.168_{\pm.007}$ | $8.468_{\pm.159}$ | $0.261_{\pm.001}$ |

Table 4: **Ablation study of three modules in MMU stage-1.** Results on HumanML3D. On unsafe sets, higher FID and lower retrieval (R@K) indicate stronger forgetting; on the safe set, lower FID and higher retrieval indicate better utility. Diversity is reported for reference.

| | **Unlearned** Unsafe Set | | | | | **Unseen** Unsafe Set | | | | | **Unseen** Safe Set | | | | |
|---|---|---|---|---|---|---|---|---|---|---|---|---|---|---|---|
| | FID$\uparrow$ | Div. | R@1$\downarrow$ | R@2$\downarrow$ | R@3$\downarrow$ | FID$\uparrow$ | Div. | R@1$\downarrow$ | R@2$\downarrow$ | R@3$\downarrow$ | FID$\downarrow$ | Div. | R@1$\uparrow$ | R@2$\uparrow$ | R@3$\uparrow$ |
| SafeMo ($\alpha = 0.0$) | 1.7197 | 7.3746 | 0.2517 | 0.3914 | 0.4969 | 2.3050 | 7.5191 | 0.2365 | 0.3896 | 0.5052 | 0.5232 | 9.3375 | 0.3755 | 0.5599 | 0.6732 |
| SafeMo-Static | 8.0235 | 6.8083 | 0.2016 | 0.3164 | 0.4043 | 9.0499 | 6.8880 | 0.1958 | 0.3167 | 0.3937 | 2.5539 | 8.7060 | 0.3172 | 0.4935 | 0.6052 |
| SafeMo-Static w/o GD | 5.2830 | 6.9963 | 0.2188 | 0.3449 | 0.4377 | 5.7963 | 6.9634 | 0.2104 | 0.3333 | 0.4208 | 1.4697 | 8.8189 | 0.3347 | 0.5144 | 0.6295 |
| SafeMo-Static w/o RD | 5.7285 | 7.1058 | 0.2195 | 0.3378 | 0.4307 | 6.7159 | 7.2363 | 0.1990 | 0.3375 | 0.4375 | 1.9663 | 9.0015 | 0.3432 | 0.5263 | 0.6379 |
| SafeMo-Static w/o PD | 8.9693 | 6.6601 | 0.1960 | 0.3092 | 0.3962 | 10.5409 | 6.7565 | 0.1896 | 0.3000 | 0.3740 | 2.9845 | 8.6333 | 0.3178 | 0.4878 | 0.6040 |
| SafeMo-Gated | 28.0806 | 5.0169 | 0.0947 | 0.1630 | 0.2168 | 28.0574 | 4.8520 | 0.0865 | 0.1542 | 0.2104 | 0.5355 | 9.3224 | 0.3775 | 0.5628 | 0.6769 |
| SafeMo-Gated w/o GD | 46.9030 | 2.8490 | 0.0544 | 0.1055 | 0.1543 | 45.4955 | 2.7078 | 0.0615 | 0.1083 | 0.1542 | 0.5248 | 9.3258 | 0.3760 | 0.5624 | 0.6768 |
| SafeMo-Gated w/o RD | 21.3313 | 5.6771 | 0.1449 | 0.2351 | 0.3002 | 21.3717 | 5.5564 | 0.1469 | 0.2292 | 0.2760 | 0.5385 | 9.3204 | 0.3775 | 0.5625 | 0.6742 |
| SafeMo-Gated w/o PD | 31.0764 | 4.7098 | 0.0926 | 0.1497 | 0.2020 | 31.3418 | 4.5750 | 0.1042 | 0.1688 | 0.2073 | 0.5380 | 9.3241 | 0.3783 | 0.5631 | 0.6761 |

forget-set performance closer to models trained on a toxicity-free dataset is better. On retain-set, the comparison remains the same, where lower FID and higher retrieval is better. In short, in this work, we consider that a method that has low-quality performance on forget-set, but also demonstrates highly-maintained good performance on retain-set, is better.

**Quantitative results.** We design two deployment regimes. SafeMo-Static uses a fixed scaling factor $\alpha = 1.0$ for all prompts and requires no external classifer. SafeMo-Gated uses the SafeMo-Engine toxicity classifier to apply $\alpha = 2.0$ to unsafe prompts for forget-set and $\alpha = 0.05$ to benign prompts for retain-set. Results on HumanML3D are shown in Table 2. SafeMo-Static attains retain-set R@1 0.335, surpassing MoMask w/ LCR (0.287, +16.7%) and BAMM w/ LCR (0.299, +12.0%), while outperforming BAMM w/ LCR on forget-set FID (+5.9%) and remaining comparable to MoMask w/ LCR. SafeMo-Gated further strengthens unlearning on the forget set, with FID increases of +150.5% and +220.7% and R@1 drops of -35.3% and -36.8% relative to MoMask w/ LCR and BAMM w/ LCR, respectively, while demonstrating high retain-set quality (R@1 0.386; +34.5% vs. MoMask w/ LCR and +29.1% vs. BAMM w/ LCR) with comparable FID as a continuous and diffusion-based model. The same trend holds on Motion-X, as results shown in Table 3. SafeMo-Static and SafeMo-Gated yield forget-set FID increases of +372.8% and +1344.5%, with R@1 degradations of -48.4% and -73.5% vs. MoMask w/ LCR. On the retain set, SafeMo-Gated attains a lower FID (-56.0%), while SafeMo-Static remains comparable. In summary, SafeMo-Gated, with prompt-toxicity-awareness, has strong forgetting capability on unsafe prompts while maintaining high fidelity on the benign prompts, while SafeMo-Static acts as an external-agent-free, offline variant still pushing the unsafe generative distribution away at a comparable level with modest degradation on retain quality. On the forget split, our model exhibits neutralization: FID substantially increases and R@1 sharply drops, indicating effective removal of unsafe semantics. Meanwhile, crucially, on the retain set, FID and R@1 remain at a comparable level with base model's result, largely preserving normal utility. Comprehensive ablation studies are provided in Appendix D.

**Qualitative results.** Figure 4 compares SafeMo-Static and SafeMo-Gated on unsafe and safe prompts, illustrating stronger forgetting on unsafe intents and preserved fidelity on benign prompts. More qualitative results, additional examples, and discussion can be found in Appendix C.

Figure 4: Qualitative results of our models.

## 4.4 ABLATION STUDY

We ablate the three stage-1 modules in MMU to assess their roles in the safety-utility tradeoff. Results are reported in Table 4. For the gated setting, we use the LLM-based prompt classfier in SafeMoEngine to determine toxicity at inference time. Results are shown in Table 4. Following SKU (Liu et al., 2024b), we disentangle in-distribution forgetting and its generalization by evaluating on both unlearned and unseen unsafe prompts, while measuring benign utility on unseen safe prompts. Removing Guided Distortion (GD) substantially weakens forgetting in the static regime. The unsafe-set FID drops from 8.0235 to 5.2830 (-34.2%) on the unlearned unsafe set, and from 9.0499 to 5.7963 (-36.0%) on the unseen unsafe set, indicating that GD is a primary contributor to capturing harmful motion patterns during stage-1 editing. In contrast, for SafeMo-Gated, removing GD increases unsafe FID (from 28.0806 to 46.9030 on unlearned unsafe set, and from 28.0574 to 45.4955 on unseen unsafe set) while yielding essentially unchanged benign performance, suggesting that toxicity-aware scaling can partially compensate and may even amplify forgetting when the edited direction becomes less constrained. We therefore keep GD to obtain a stable trade-off across both deployment regimes. Without Random Decoupling (RD), unsafe-set FID decreases from 8.0235 to 5.7285 (-28.6%) on the unlearned unsafe set and from 9.0499 to 6.7159 (-25.8%) on the unseen unsafe set for SafeMo-Static; similarly, SafeMo-Gated's FID drops on both unsafe sets. Meanwhile, removing RD slightly improves benign utility for SafeMo-Static, highlighting RD as a key factor that strengthens forgetting at the cost of some utility. Preservation Divergence (PD) mainly stabilizes benign utility in the static regime. Removing PD worsens the benign performance, increasing retain-set FID from 2.5539 to 2.9845 (+16.9%) and reduces higher-K retrieval, while the effect on the gated regime is marginal. This indicates that PD is beneficial when no external toxicity-aware gating is available. Additional ablations on loss-terms, LoRA rank, and alpha-scaling are provided in Appendix D.

## 5 CONCLUSION

In this work, we introduce an innovative continuous domain-based human motion unlearned generative model, SafeMo, for trustworthy motion generation. We introduce the first safe text-to-motion dataset, SafeMoVAE-29K, along with its discrete version, to facilitate future research and standardized benchmarking in human motion unlearning. The proposed absorb-then-negate machine unlearning strategy designed for text-to-motion models, Minimal Motion Unlearning, enables selective knowledge unlearning on unsafe motions while preserving benign task performance on safe prompts. Extensive experiments on HumanML3D and Motion-X datasets demonstrate that our model achieves SOTA performance on human motion unlearning.

## 6 LIMITATIONS

To our knowledge, this work is among the first to study human motion unlearning on a continuous latent-space. On unsafe prompts, our goal is *semantic removal*, i.e., preventing the model from expressing the unsafe motion semantics, rather than producing a high-fidelity safe substitute motion. However, this safety-first operating point may lead to over-suppression for some unsafe prompts, e.g., stationary-like patterns, and can exacerbate kinematic artifacts such as foot-skating, especially under larger negation gating scales. Additional discussions on failure modes and future work are provided in Appendix F.

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

## A  MINIMAL MOTION UNLEARNING (MMU) ALGORITHM

---

**Algorithm 1** Minimal Motion Unlearning (MMU)

---

**Require:** Unsafe set $\mathcal{U}$, Safe set $\mathcal{S}$; base DiP model $f_{\theta_0}$; LoRA config $(r, \alpha_{\text{LoRA}}, p_{\text{dropout}})$; loss weights $\{\lambda, \mu, \beta\}$; diffusion schedule.
**Ensure:** Task vector $\Delta\theta$ and safe model $f_{\theta^*}$
 1: Initialize $\theta \leftarrow \theta_0$; insert LoRA adapters at attention/FFN; freeze non-LoRA params.

 2: **Stage 1: Harmful Knowledge Absorption (training)**
 3: **repeat**
 4:      Sample unsafe batch $\mathcal{U}_b$, safe batch $\mathcal{S}_b$, timesteps $t$, noise $\varepsilon$.
 5:      **for** $i \in \mathcal{U}_b$ **do**
 6:          Generate noisy $x_t^{(i)} \sim q(x_0^{(i)}, t, \varepsilon)$
 7:          Predict $\hat{x}_0^{(i)} \leftarrow f_\theta(x_t^{(i)}, t, C^{(i)})$
 8:          Compute $\mathcal{L}_{\text{harm}}^{(i)}$ using Eq. 1
 9:          Build decoupled $\tilde{x}_{\text{tgt}}^{(i)}$ (segment shuffle / reverse)
10:          Sync prefix $\tilde{C}^{(i)} \leftarrow \text{SyncPrefix}(C^{(i)}, \tilde{x}_{\text{tgt}}^{(i)})$
11:          Predict $\hat{x}_0^{\text{mix}} \leftarrow f_\theta(x_t^{(i)}, t, \tilde{C}^{(i)})$
12:          Compute $\mathcal{L}_{\text{dec}}^{(i)}$ via Eq. 3
13:      **end for**
14:      **for** $j \in \mathcal{S}_b$ **do**
15:          Generate $x_t^{(j)} \sim q(x_0^{(j)}, t, \varepsilon)$
16:          Extract pooled features $z_{\text{cur}}^{(j)}, z_{\text{base}}^{(j)}$
17:          Create decoupled $\tilde{x}_0^{(j)}$ and synced prefix $C^{\text{dec}}$
18:          Extract decoupled pooled features $z_{\text{cur}}^{\text{dec}}, z_{\text{base}}^{\text{dec}}$
19:          Compute $\mathcal{L}_{\text{pres}}^{(j)}$ via Eq. 4
20:      **end for**
21:      Form stage-1 objective $\mathcal{L}_{\text{Stage1}}$ by Eq. 5
22:      Update only LoRA parameters with $\nabla_\theta \mathcal{L}_{\text{Stage1}}$
23: **until** convergence
24: $\theta_{\text{harm}} \leftarrow \theta, \Delta\theta \leftarrow \theta_{\text{harm}} - \theta_0$

25: **Stage 2: Harmful Knowledge Negation (inference)**
26: **for** prompt (text, class) **do**
27:      **if Static then**
28:          $\alpha \leftarrow \alpha_{\text{Static}}$
29:      **else if Gated then**
30:          **if** class = `unsafe` **then** $\alpha \leftarrow \alpha_{\text{unsafe}}$ **else** $\alpha \leftarrow \alpha_{\text{safe}}$
31:      **end if**
32:      Set $\theta^* \leftarrow \theta_0 - \alpha \Delta\theta$
33:      Generate motion via DDPM-style denoising with $f_{\theta^*}$
34: **end for**

---

## B  EVALUATION METRICS

We evaluate with three standard metrics: R-Precision (R@k), Fréchet Inception Distance (FID), and Diversity. Below we give brief definitions.

**R-Precision.**  Following t2m (Zhang et al., 2023), a shared text–motion feature space is used for retrieval. R-Precision reports top-$k$ accuracy when each generated motion is queried against 1 ground-truth caption and 31 mismatched captions (R@1/2/3).

**FID.** FID computes the Fréchet distance between two Gaussians fitted to motion features from generated and real samples, capturing distributional discrepancy. This is measured by the L2-loss between their latent feature representations.

**Diversity.** Diversity estimates intra-set variability by splitting the generated set into two equal halves $\{m_1, \ldots, m_{M_d}\}$ and $\{m'_1, \ldots, m'_{M_d}\}$ and averaging cross-set feature distances:

$$\text{Diversity} = \frac{1}{M_d} \sum_{i=1}^{M_d} \|m_i - m'_i\|_2. \tag{7}$$

## C  QUALITATIVE RESULTS

We present results on unsafe prompts in Figure 5 and Figure 6, and results on safe prompts in Figure 7.

From the results on unsafe prompts in Figure 5 and Figure 6, we observe that both SafeMo-Static and SafeMo-Gated can effectively erase the toxic motion semantics, which aligns with the design and aim of our unlearning strategy. SafeMo-Static erases toxic motion semantics effectively, while SafeMo-Gated tends to generate stationary or repeated pattern-like motion, which demonstrates a stronger tone of unlearning.

However, some limitations and flaws are observed in our qualitative results. Firstly, although it is of a high success rate that the model's generated results are not aligned with the unsafe text prompts, we observe some suboptimal results in certain scenarios, e.g., very long and detailed descriptions will cause some atomic semantics to be omitted, or being in a stationary-like pattern.

Secondly, we observe foot sliding and skating artifacts as a byproduct of the unlearning, with increased occurrence when applying a larger alpha to the text vector. We also observed that from Table 5, terms like $\mathcal{L}_{\text{foot}}$ are not in a linear-mapping fashion, i.e., with a larger alpha applied, the performance gaps are not changing in a linear manner. This indicates that we may need to explore a more complex relationships between the unlearning effect and each term of our designed method to further improve the model's performance in future work.

As the results shown in Figure 7, both SafeMo-Static and SafeMo-Gated can generate semantically aligned results on safe prompts. In some cases, SafeMo-Static favors lower-amplitude, more conservative kinematics. Additionally, foot-sliding and skating artifacts as a byproduct of the unlearning are also observed. SafeMo-Static is more susceptible to this byproduct than SafeMo-Gated because of the larger $\alpha$ weighted task vector negation applied on it on safe prompts than on SafeMo-Gated.

## D  ABLATION STUDY

### D.1  LOSS SUBTERM REMOVAL

We conducted ablation experiments by iteratively removing the loss terms, MPJPE, $\mathcal{L}_{\text{vel}}$, $\mathcal{L}_{\text{acc}}$, $\mathcal{L}_{\text{foot}}$, and $\mathcal{L}_{\text{text}\leftrightarrow\text{mo}}$, to demonstrate each term's significance in the model learning process. As shown in Table 5, the removal of MPJPE drastically destroys the model's unlearning performance on the unsafe set. The $\mathcal{L}_{\text{vel}}$ and $\mathcal{L}_{\text{acc}}$ both play an important role in the model's unlearning of unsafe patterns as well, with lower FID and higher R precision on unsafe prompts after removing them. $\mathcal{L}_{\text{foot}}$ plays a role in enhancing the model's understanding of the motion semantics and helps generate more physically aligned results, with slight degradation in unlearning on unsafe prompts after removing it. $\mathcal{L}_{\text{text}\leftrightarrow\text{mo}}$ has a significant influence on the model's understanding of unsafe motion since we observe lower FIDs on both versions of the model on unsafe prompts.

### D.2  LoRA RANK ABLATION

Unless otherwise specified, we use LoRA with rank $r = 16$ as a prior default, which offers a stable capacity-regularization trade-off in our decoder-only DiP backbone. To evaluate the effect of different LoRA (Hu et al., 2022) rank, we conduct an ablation study on different LoRA rank values. The results are shown in Table 6. We evaluate the same checkpoints after training on the MMU

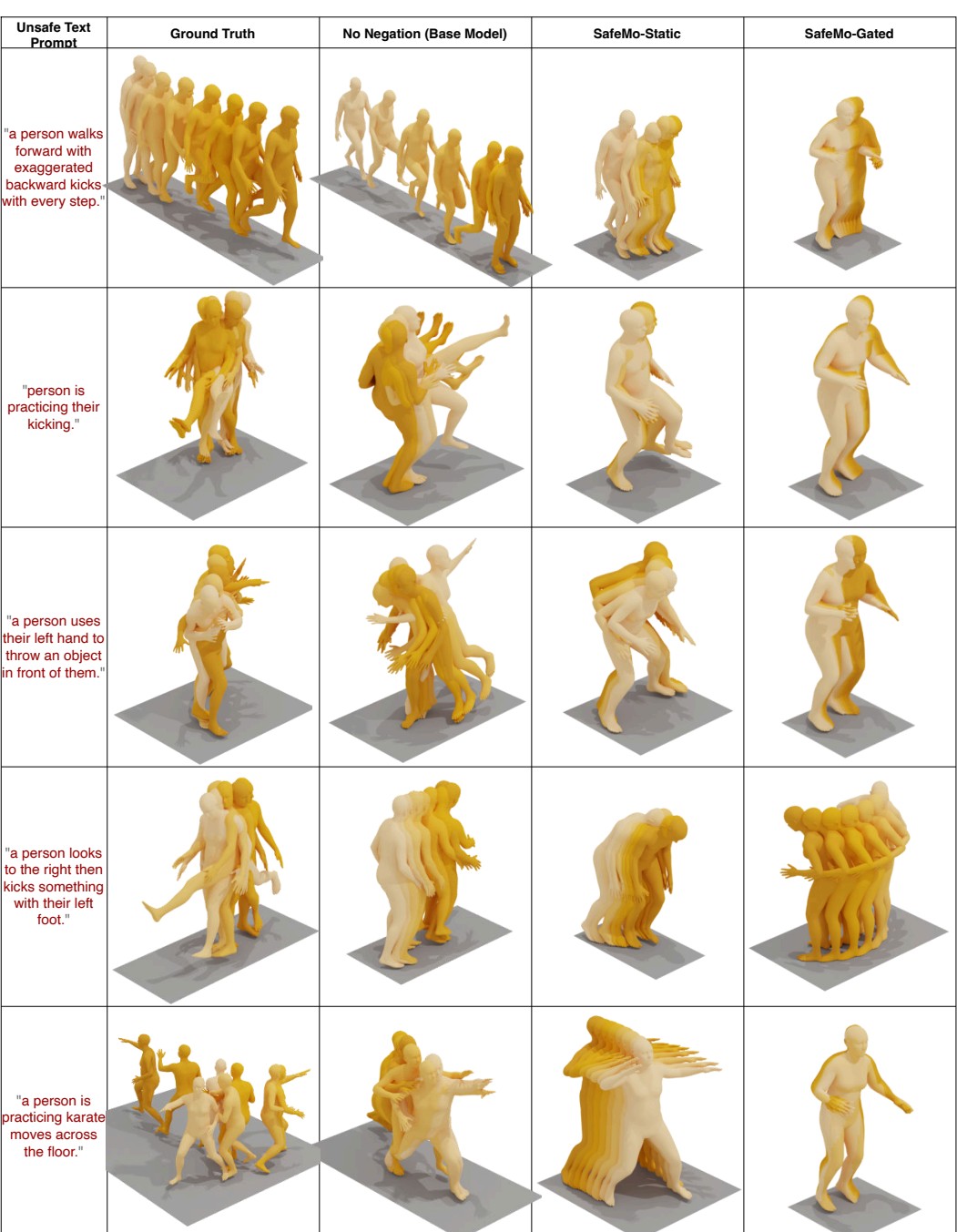

Figure 5: Qualitative results of generated motions on unsafe prompts (Part I).

strategy for 20K steps with different LoRA ranks and the same LoRA alpha as the LoRA rank, while keeping all other hyperparameters the same. Across different sets, $r = 16$ yields the most balanced forgetting-retention effect: unsafe FID increases while unsafe R@k is reduced or comparable, and performance on the safe set is maintained to an acceptable level with the best balanced results on applying the same checkpoint for SafeMo-Static and SafeMo-Gated. We hereby unfold our findings to support our choice. While $r = 8$ sometimes produces slightly higher FID shifts on unsafe subsets, it frequently exhibits higher R@k, which indicates a more sense of geometric displacement and a lower level of commensurate semantic forgetting on unsafe prompts. We also observe that with the same replication times, the confidence intervals (CI) of $r = 8$ are consistently larger than on $r = 16$,

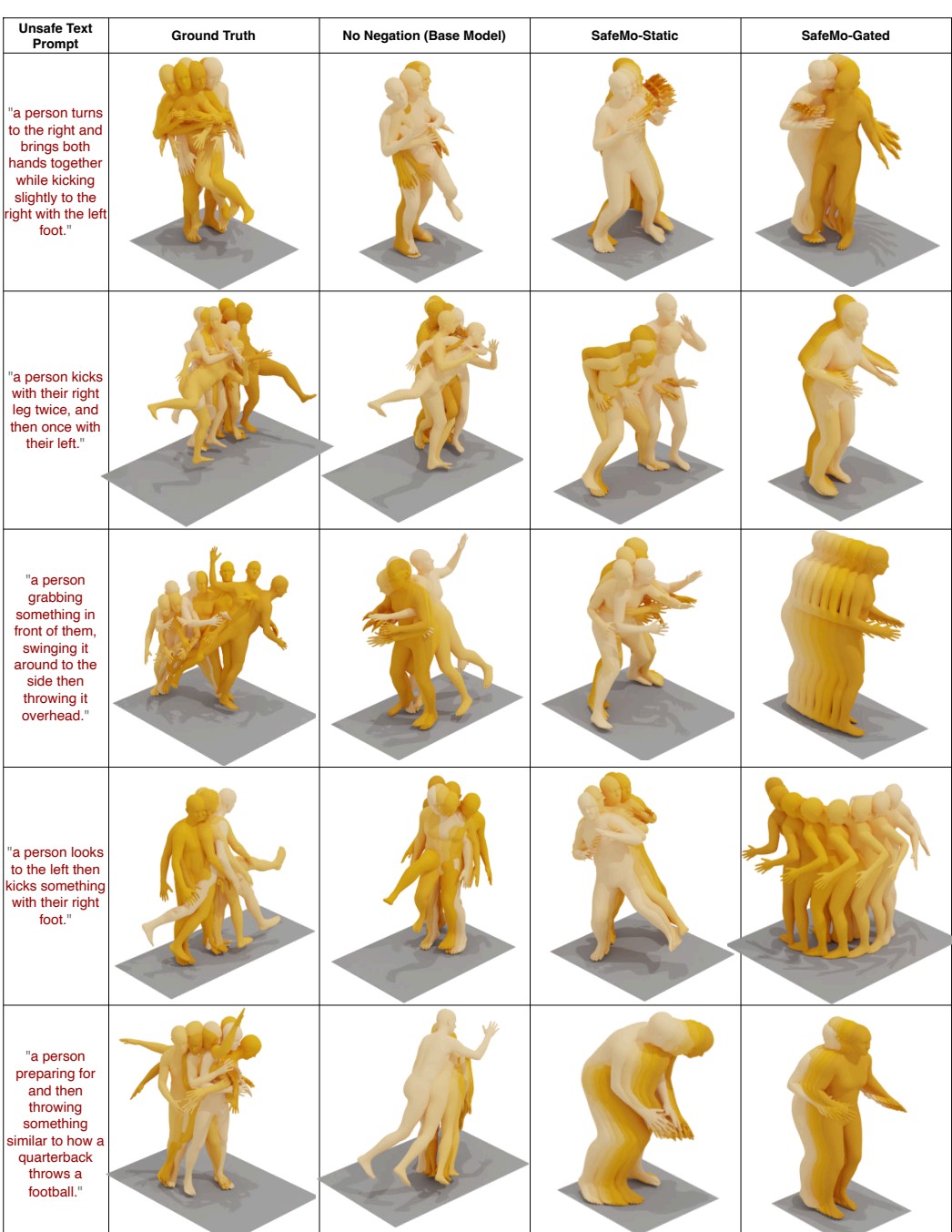

Figure 6: Qualitative results of generated motions on unsafe prompts (Part II).

which is a sign of instability and inadequate capability of obtaining the exact knowledge we want during the first stage. Conversely, model with $r = 32$ tends to under-forget on the unsafe sets with relatively large performance gaps on FID and R precisions on both the Gated and the Static settings. Apart from that, in the Gated setting, it greatly harms the safe set's performance even with a small value of $\alpha$ in the Gated setting, making it undesirable.

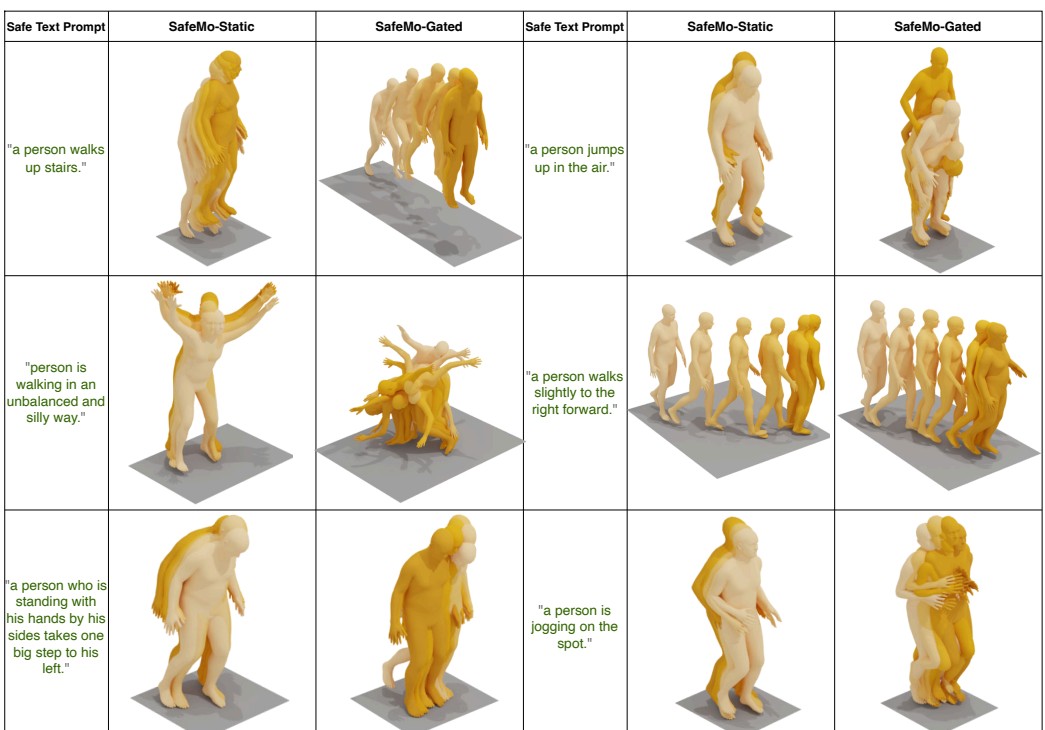

Figure 7: Qualitative comparison of generated motions on safe prompts.

Table 5: **Ablation study of loss subterms in MMU stage-1.** Results on HumanML3D. On unsafe sets, higher FID and lower retrieval (R@K) indicate stronger forgetting; on the safe set, lower FID and higher retrieval indicate better utility. Diversity is reported for reference.

| | **Unlearned** Unsafe Set | | | | | **Unseen** Unsafe Set | | | | | **Unseen** Safe Set | | | | |
|---|---|---|---|---|---|---|---|---|---|---|---|---|---|---|---|
| | FID↑ | Div. | R@1↓ | R@2↓ | R@3↓ | FID↑ | Div. | R@1↓ | R@2↓ | R@3↓ | FID↓ | Div. | R@1↑ | R@2↑ | R@3↑ |
| SafeMo ($\alpha = 0.0$) | 1.7197 | 7.3746 | 0.2517 | 0.3914 | 0.4969 | 2.3050 | 7.5191 | 0.2365 | 0.3896 | 0.5052 | 0.5232 | 9.3375 | 0.3755 | 0.5599 | 0.6732 |
| SafeMo-Static | 8.0235 | 6.8083 | 0.2016 | 0.3164 | 0.4043 | 9.0499 | 6.8880 | 0.1958 | 0.3167 | 0.3937 | 2.5539 | 8.7060 | 0.3172 | 0.4935 | 0.6052 |
| SafeMo-Static w/o MPJPE | 4.6513 | 7.3699 | 0.2243 | 0.3654 | 0.4587 | 4.7538 | 7.3980 | 0.2271 | 0.3729 | 0.4677 | 1.7615 | 9.0878 | 0.3548 | 0.5368 | 0.6493 |
| SafeMo-Static w/o $\mathcal{L}_{vel}$ | 6.9403 | 6.9088 | 0.2108 | 0.3318 | 0.4221 | 7.8048 | 7.0443 | 0.1885 | 0.3229 | 0.4167 | 2.1361 | 8.8828 | 0.3346 | 0.5123 | 0.6286 |
| SafeMo-Static w/o $\mathcal{L}_{acc}$ | 7.7847 | 6.8950 | 0.2074 | 0.3266 | 0.4147 | 8.8772 | 6.9484 | 0.3073 | 0.4479 | 0.5434 | 2.3508 | 8.8585 | 0.3266 | 0.5078 | 0.6199 |
| SafeMo-Static w/o $\mathcal{L}_{foot}$ | 7.5536 | 6.8465 | 0.2031 | 0.3260 | 0.4109 | 8.5371 | 6.9620 | 0.1854 | 0.3198 | 0.4094 | 2.3039 | 8.7675 | 0.3262 | 0.5037 | 0.6139 |
| SafeMo-Static w/o $\mathcal{L}_{text\leftrightarrow mo}$ | 7.0224 | 6.7054 | 0.2039 | 0.3295 | 0.4172 | 7.9530 | 6.7763 | 0.1979 | 0.3125 | 0.4094 | 2.2494 | 8.6139 | 0.3253 | 0.5086 | 0.6201 |
| SafeMo-Gated | 28.0806 | 5.0169 | 0.0947 | 0.1630 | 0.2168 | 28.0574 | 4.8520 | 0.0865 | 0.1542 | 0.2104 | 0.5355 | 9.3224 | 0.3775 | 0.5628 | 0.6769 |
| SafeMo-Gated w/o MPJPE | 11.7242 | 6.8297 | 0.1962 | 0.3154 | 0.4026 | 11.9482 | 6.9063 | 0.2115 | 0.3167 | 0.4052 | 0.5600 | 9.3109 | 0.3743 | 0.5611 | 0.6761 |
| SafeMo-Gated w/o $\mathcal{L}_{vel}$ | 25.0236 | 5.1915 | 0.1238 | 0.2037 | 0.2600 | 25.4520 | 5.1998 | 0.1292 | 0.2167 | 0.2875 | 0.5330 | 9.4388 | 0.3713 | 0.5641 | 0.6751 |
| SafeMo-Gated w/o $\mathcal{L}_{acc}$ | 25.2587 | 5.1204 | 0.1252 | 0.2076 | 0.2681 | 25.3217 | 5.1177 | 0.1229 | 0.2125 | 0.2771 | 0.5327 | 9.4913 | 0.3814 | 0.5701 | 0.6802 |
| SafeMo-Gated w/o $\mathcal{L}_{foot}$ | 27.1904 | 4.9923 | 0.1152 | 0.1858 | 0.2411 | 27.6051 | 4.9489 | 0.1208 | 0.2021 | 0.2615 | 0.5327 | 9.4370 | 0.3719 | 0.5649 | 0.6768 |
| SafeMo-Gated w/o $\mathcal{L}_{text\leftrightarrow mo}$ | 22.3646 | 4.7543 | 0.1076 | 0.1825 | 0.2429 | 22.7699 | 4.7297 | 0.1031 | 0.1948 | 0.2521 | 0.5263 | 9.4303 | 0.3726 | 0.5659 | 0.6764 |

## D.3 ALPHA SCALING ABLATION

We study how the task-vector scale $\alpha$ controls the trade-off between retaining benign capability and forgetting harmful behaviors. The results are shown in Figure 8. The curves reveal a clear effective-unlearning window on $[0.05, 1.2]$, where the unsafe split deteriorates markedly with mild changes happening on safe prompts. Beyond $\alpha = 1.2$, the safe curves also degrade steeply indicating over-forgetting, which is undesirable for benign prompts. We reckon that $\alpha = 1.0$ is the sweet spot for SafeMo-Static, with acceptable degradation on benign tasks (FID = 2.51, R@1 = 0.33) and good unlearning performance on unsafe prompts (FID = 9.55, R@1 = 0.18). These observations demonstrate the large selective deterioration on unsafe prompts before the knee, validating that our unlearning is effective. These also motivate the SafeMo-Gated setting for a more flexible and accurate control utilizing the proposed LLM-base agent in *SafeMoEngine*.

Table 6: **Ablation study of LoRA rank in MMU stage-1.** Results on HumanML3D. On unsafe sets, higher FID and lower retrieval (R@K) indicate stronger forgetting; on the safe set, lower FID and higher retrieval indicate better utility. Diversity is reported for reference.

| | **Unlearned** Unsafe Set | | | | | **Unseen** Unsafe Set | | | | | **Unseen** Safe Set | | | | |
|---|---|---|---|---|---|---|---|---|---|---|---|---|---|---|---|
| | FID↑ | Div. | R@1↓ | R@2↓ | R@3↓ | FID↑ | Div. | R@1↓ | R@2↓ | R@3↓ | FID↓ | Div. | R@1↑ | R@2↑ | R@3↑ |
| SafeMo ($\alpha = 0.0$) | 1.7197 | 7.3746 | 0.2517 | 0.3914 | 0.4969 | 2.3050 | 7.5191 | 0.2365 | 0.3896 | 0.5052 | 0.5232 | 9.3375 | 0.3755 | 0.5599 | 0.6732 |
| SafeMo-Static | 8.0235 | 6.8083 | 0.2016 | 0.3164 | 0.4043 | 9.0499 | 6.8880 | 0.1958 | 0.3167 | 0.3937 | 2.5539 | 8.7060 | 0.3172 | 0.4935 | 0.6052 |
| SafeMo-Static (LoRA r=8) | 8.1880 | 6.8448 | 0.2022 | 0.3225 | 0.4117 | 8.9847 | 6.9679 | 0.1958 | 0.3250 | 0.4042 | 2.4408 | 8.6262 | 0.3252 | 0.5031 | 0.6127 |
| SafeMo-Static (LoRA r=32) | 5.6007 | 7.7033 | 0.2068 | 0.3372 | 0.4325 | 4.9216 | 7.6619 | 0.1854 | 0.3333 | 0.4396 | 2.4861 | 8.7952 | 0.2947 | 0.4700 | 0.5841 |
| SafeMo-Gated | 28.0806 | 5.0169 | 0.0947 | 0.1630 | 0.2168 | 28.0574 | 4.8520 | 0.0865 | 0.1542 | 0.2104 | 0.5355 | 9.3224 | 0.3775 | 0.5628 | 0.6769 |
| SafeMo-Gated (LoRA r=8) | 31.5091 | 4.1975 | 0.1080 | 0.1800 | 0.2313 | 32.7289 | 4.1776 | 0.1052 | 0.1750 | 0.2417 | 0.5328 | 9.2650 | 0.3781 | 0.5666 | 0.6785 |
| SafeMo-Gated (LoRA r=32) | 19.6348 | 6.7620 | 0.1292 | 0.2184 | 0.2924 | 17.7689 | 6.7304 | 0.1365 | 0.2344 | 0.3052 | 2.5434 | 9.0249 | 0.3222 | 0.4983 | 0.6139 |

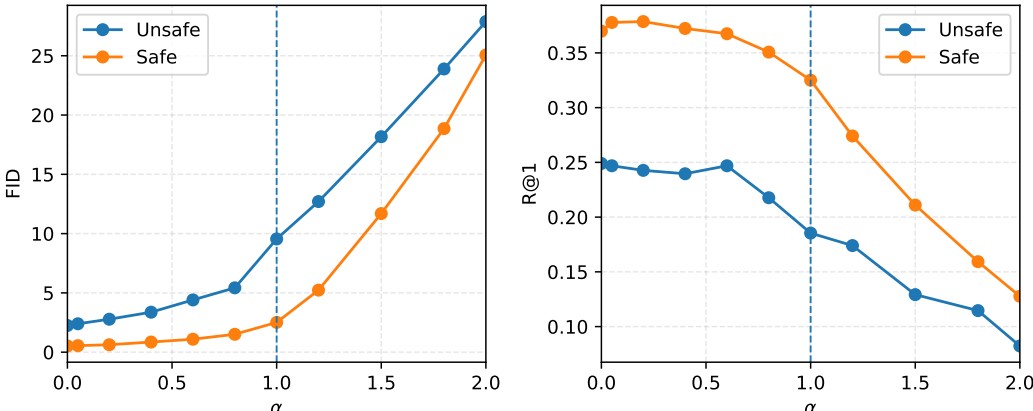

Figure 8: **Effect of the task-vector scaling $\alpha$ on safe and unsafe prompts.** Left: FID (lower is better on safe prompts; higher indicates stronger forgetting on unsafe prompts). Right: R@1 (higher is better on safe prompts; lower indicates stronger forgetting on unsafe prompts). The dashed line marks $\alpha = 1.0$.

## E    IMPLEMENTATION DETAILS

**Forget and retain set partitioning.**    To compare our method with LCR (De Matteis et al., 2025) despite the fact that they have not made their implementation open-source, we adopt the same keyword-based partitioning paradigm they describe. We construct a curated list of harmful action lemmas as described in their method, lemmatize captions, and perform exact lemma-level matching with phrase-first priority. A sample is assigned to the *forget set* if *any* of its captions hits the list; otherwise, it belongs to the *retain set*.

**Motion-X.**    Motion-X (Lin et al., 2023) provides SMPL-X (Pavlakos et al., 2019) data including hand and facial feature, which is not aligned with our setup. We process it using the official code from Motion-X (Lin et al., 2023), converting SMPL-X features to SMPL (Loper et al., 2015) representations. To ensure the comparability with LCR, we preprocess the text prompts using HumanML3D (Zhang et al., 2023)'s Semantic Role Labeling method, and train feature extractors following official t2m (Zhang et al., 2023) implementation for 300 epochs, as the same as in LCR.

## F    LIMITATION AND FUTURE WORK

This appendix expands the brief limitations stated in the main paper, with concrete failure cases and future directions for both SafeMoEngine and MMU.

**Safe T2M dataset.**    Despite that we design the LLM-based agent in a classify-then-rewrite fashion with explicit few-shot, rule-enhanced prompt engineering, and effective, level-aligned rewriting strategy, we acknowledge several limitations. Firstly, we have observed that the classifier agent tends to be slightly oversensitive to toxic semantics. In level-2 prompts, we found a few sport-related or dancing-related prompts, e.g., golf or dancing with crazy legs. Secondly, although we apply two recent advances, one continuous and the other discrete token-based, for generating new

refined safe motion for unsafe prompts, some generated results can be suboptimal due to the base model's respective limitations.

**Minimal motion unlearning.** Although our results demonstrate the effectiveness of unsafe motion unlearning, we have found several kinds of suboptimal cases or failed cases. First, when the context is too long and contains many details, the generated motion can omit some atomic semantics, or even, at worst, become a stationary-like pattern. We reckon that this is the base model's limited understanding capability of long and complex prompts. In future work, we aim to address this problem by using a more semantic-accurate base model or designing a text prompt reasoning helper, e.g., methods similar to Motion-agent (Wu et al., 2024). Secondly, task-vector negation can exacerbate foot-skating artifacts, which are particularly noticeable for SafeMo-Gated under larger gating scales. Future work may mitigate this byproduct by designing more physics-guided constraints and integrating physics-based trackers, such as in CLoSD (Tevet et al., 2024), to further improve physically plausible contacts and environment interactions. Thirdly, Operating in continuous motion space alleviates discrete codebook stitching artifacts (Figure 1), but it may trade off some standard T2M fidelity metrics compared to strong VQ-token pipelines.

## G  USER STUDY

We conduct a comprehensive user study to evaluate the overall quality and unlearning performance of motion sequences generated by our methods. A total of 50 participants completed a Google Forms survey designed to assess the physical plausibility, unlearning outcomes on unsafe prompts, and benign performance on safe prompts.

As illustrated in Figure 9, section 1 and 2 displays different generated results on unsafe prompts by SafeMo-Static and SafeMo-Gated respectively, followed by section 3 and 4 showing SafeMo-Gated and SafeMo-Static's generated results on safe prompts, with 3 different prompts and 2 different questions each section. Section 5 and 6 then compare SafeMo-Static and SafeMo-Gated's results on the same safe text prompt and the same unsafe prompt respectively. Participants are asked to rate each motion at a 5-point Likert scale (where 1 represents low and 5 represents high) based on motion naturalness, degree of unlearning performance on unsafe prompts and quality of text alignment on safe prompts.

This study aims to evaluate not only the unlearning performance of our model, but also the benign performance and quality distinctions between SafeMo-Static and SafeMo-Gated.

The results of the user study can be summarized as follows: *(i)* Our method achieved an overall motion quality of 4.24 on SafeMo-Static, and 4.82 on SafeMo-Gated. *(ii)* On unsafe prompts, 98% of participants agreed that SafeMo-Gated effectively removed unsafe components in the motion, and 86% on SafeMo-Static. *(iii)* On safe prompts, our method's generated results on safe set scored 4.64 on text-motion visual alignment rating. *(iv)* 56% of the participants preferred SafeMo-Gated method on unsafe prompts, and 84% preferred SafeMo-Gated method on safe prompts.

## H  LLM USE DECLARATION

Large Language Models (ChatGPT) were used exclusively to improve the clarity and fluency of English writing. They were not involved in research ideation, experimental design, data analysis, or interpretation. The authors take full responsibility for all content.

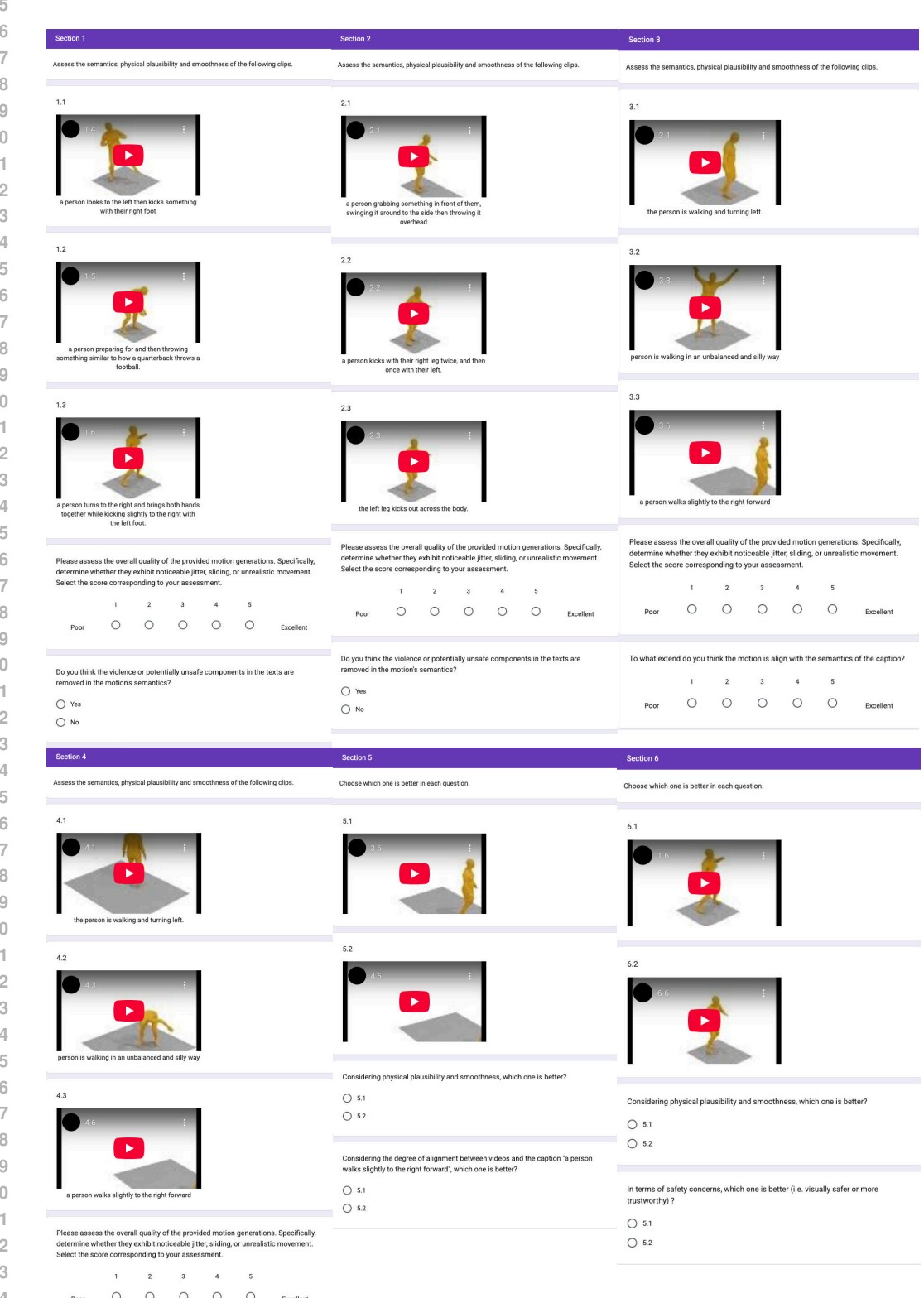

Figure 9: **User study Google Forms.** The User Interface (UI) used in our user study.

