# OpenReview forum: "SafeMo: Trustworthy Motion Generation"
_ICLR.cc/2026/Conference — ICLR 2026 Conference Withdrawn Submission_

### Official Review · Reviewer_axJz · 2025-10-28

**Soundness:** 3
**Presentation:** 3
**Contribution:** 2
**Rating:** 4
**Confidence:** 3

**Summary:**

This paper proposes SafeMo, a trustworthy text-to-motion generative framework equipped with a two-stage Minimal Motion Unlearning (MMU) method that selectively removes unsafe behaviors while preserving benign generation ability. It also introduces SafeMoVAE-29K, the first safety-oriented text-to-motion dataset (and its discrete version, SafeMoVQ-29K), providing a standardized benchmark for safe motion generation and unlearning.

**Strengths:**

1.	The exploration of trustworthy motion generation is novel, as this topic has been rarely studied in motion generation tasks.
2.	The optimization of textual annotations in the HumanML3D dataset represents a significant engineering contribution.

**Weaknesses:**

1.	Many technical details remain unclear. For example, how does the LLM rewrite the text prompts in HumanML3D? What specific prompts are used? What does the overall model architecture look like?
2.	As a new task in motion generation, this work aims to establish a new benchmark. However, as shown in Table 2, the benchmark lacks sufficient discussion. Specific issues include: a. In the comparison with existing methods, are there more recent or representative approaches beyond MoMask and BAMM that could better highlight the effectiveness of the proposed strategy? b. What is the meaning of the right arrow after each metric, and why are the evaluation criteria different between the Forget Set and Retain Set?
3.	As I understand it, the proposed method aims to improve motion generation quality by reducing harmful inputs. Therefore, the model’s effectiveness should also be validated on conventional text-to-motion tasks.
4.	Similarly, the visualization experiments only include ablation studies comparing internal variants of the proposed method, lacking comparisons with current state-of-the-art approaches.

**Questions:**

Please refer to the weaknesses above.

---

### Official Review · Reviewer_WWHG · 2025-10-29

**Soundness:** 2
**Presentation:** 2
**Contribution:** 2
**Rating:** 2
**Confidence:** 4

**Summary:**

The paper presents SafeMo, a method for text2motion unlearning. It aims to build text2motion models that can avoid generating harmful content, i.e., violent motion. It first points out the limitation of vq codebook-based motion unlearning, which results in artifacts and jerky transitions. To work on a continuous space model, it adapts a knowledge unlearning method, Selective Knowledge Negation Unlearning, from LLMs, and applies it to the Diffusion Planner (DiP) model. Empirical results show that the proposed method achieved better performance on unlearning harmful content and preserving benign samples.

**Strengths:**

1. The paper focuses on a relatively new task of human motion unlearning, which is interesting.
2. It also proposes one of the first safe text-to-motion datasets, which might be useful for future results. However, the quality of the dataset has not been justified in the paper.
3. The quantitative results show that the proposed method can achieve better performance compared to the baselines.
4. The Selective Knowledge Negation Unlearning method borrowed from the language area is interesting.

**Weaknesses:**

1. One of my major concerns is that the results are not convincing. The video demo exhibits extensive artifacts, mostly just being frozen all the way, over smooth, or jittery transitions for the unsafe sample generated by the proposed method. This contradicts the user study results, which claim high quality. The video results are also much worse than the results originally presented by the baseline method [1].
2. The design of L_pre that encourages the model to diverge from the frozen baseline snapshot using "temporal-averages joint features" is problematic. Unlike the negative KL divergence used in the original method, this negative L2 distance in the temporal-averaged space doesn't make similar effects.
3. To perform the Harmful knowledge negation at test-time that blends the network parameters, I suppose that it has a very strong assumption that the interpolation of network parameters is still valid and functional. I'm a bit skeptical of this method on the relatively small motion generation models, given the poor results shown by video demos
4. The notation is hard to follow. For example, for the network parameter $\theta_{safe}$ and $\theta^*$ refer to the same model, and so do  $\theta_{bad}$ and $\theta_{harm}$. It needs more effort to polish the writing.
5. There's no description of the comparison method, such as $D_r$, $FT$, $UCE$, $RECE$. Then I found that all comparison results of the baselines are copied from [1], which raises concerns.

[1] De Matteis, Edoardo, et al. "Human Motion Unlearning." arXiv preprint arXiv:2503.18674 (2025).

**Questions:**

1. I wonder if the proposed dataset is used in the proposed method? How's the quality of the proposed dataset? I have concern on the quality since it's synthesized by generation models.
2. The complex loss terms in Equation 2 seem somewhat over-designed. I wonder whether all of these terms are necessary and effective. How important is each term in influencing the unlearning performance?

**Details Of Ethics Concerns:**

There's no description of the comparison method, such as $D_r$, $FT$, $UCE$, $RECE$. Then I found that all comparison results of the baselines are copied from [1], which raises concerns. The video results are of low quality, which seems to contradict the user study and the metrics reported.

[1] De Matteis, Edoardo, et al. "Human Motion Unlearning." arXiv preprint arXiv:2503.18674 (2025).

---

> ### Author Response · Authors · 2025-11-28
> **Regarding the research integrity concern**
>
> Thank you for the detailed feedback. We address your key concerns below.
>
> (1) Regarding the research integrity concern.
> We apologize that our original submission did not make it sufficiently explicit that several baseline numbers were reported from De Matteis et al. (2025). This was due to the unavailability of their implementations/checkpoints at the time of submission (including LCR and the motion adaptations of UCE/RECE), although we cited De Matteis et al. (2025) multiple times and provided additional details in the Appendix.
> We have fixed this transparency issue in the revised manuscript: (i) Sec. 4.3 now explicitly states that we report the corresponding baseline results from De Matteis et al. (2025) and follow their keyword-based split protocol; (ii) Tables 2–3 mark these entries with $\dagger$ and add an explicit note that $\dagger$ rows are reported from De Matteis et al. (2025).
> All results for our proposed methods (SafeMo variants and ablations) are obtained by us under our evaluation pipeline. We apologize for the earlier ambiguity. There is no plagiarism or dual submission involved.
>
> (2) “Low-quality videos” vs. user study / metrics.
> We agree that for some unsafe prompts, outputs can appear conservative (e.g., low-amplitude or stationary-like) and may exhibit kinematic artifacts (e.g., foot-skating), especially under stronger negation/gating. We explicitly acknowledge these failure modes and discuss mitigation directions (e.g., physics-guided constraints/trackers) in Appendix F (Limitations).
> Importantly, our objective on unsafe prompts is semantic removal/neutralization, preventing the model from expressing unsafe motion semantics—rather than generating a high-fidelity “safe substitute” motion. Accordingly, our user study separates: (i) motion naturalness, (ii) whether unsafe components are removed on unsafe prompts, and (iii) text–motion alignment on safe prompts. The results reflect this intended trade-off: 98% of participants agreed SafeMo-Gated removed unsafe components (86% for SafeMo-Static), while preference on unsafe prompts is 56%, indicating that over-suppression is noticeable. Meanwhile, on safe prompts, alignment remains strong (4.64/5), supporting preserved benign utility.
>
> (3) On the design of the preservation term and the loss complexity. The negative-KL used in the original LLM setting is not directly applicable to continuous motion generation; we therefore use a motion-representation-based preservation divergence (computed on pooled/temporal-averaged joint features) as a stable proxy, and we provide both module ablations and loss-term ablations in the Appendix that isolate the contribution of each component.
>
> (4) Notation clarity.
> We agree the notation can be improved. In the revision, we further unify the symbols for the base model, edited model, and task vector (and remove duplicated notations) to make the derivation easier to follow.
>
> (5) Is the proposed dataset used, and what about its quality?
> The method itself uses the original mixed dataset with a retain/forget split, while SafeMoEngine from the dataset curation pipeline is used in SafeMo-Gated as the prompt classifier. The released curated dataset is an auxiliary contribution, and we provide its construction details while acknowledging synthesis limitations.

---

### Official Review · Reviewer_4EVa · 2025-10-31

**Soundness:** 2
**Presentation:** 3
**Contribution:** 2
**Rating:** 4
**Confidence:** 3

**Summary:**

This paper introduces SafeMo, a framework for trustworthy text-to-motion (T2M) via a two-stage Minimal Motion Unlearning (MMU) procedure on a DiP diffusion backbone and a new “safe” text–motion dataset (SafeMoVAE-29K / SafeMoVQ-29K). Stage-1 absorbs “harmful” capability into LoRA adapters using a motion-aware loss (MPJPE/vel/acc/foot + text–motion alignment) plus a random decoupling trick; Stage-2 negates the learned task vector at inference by subtracting it (fixed or classifier-gated α) to suppress unsafe content while preserving benign behavior. The dataset is synthesized by an LLM agent that classifies prompts into safe/risky/unsafe and rewrites the latter, then regenerates motions with MotionFlow Transformer (continuous) and MotionAgent (discrete). On HumanML3D and Motion-X, SafeMo increases forget-set FID and lowers R@1 more than LCR, while retaining competitive or better benign quality (notably with the gated variant).

**Strengths:**

-  LoRA-only updates with vector negation (static/gated α) enable plug-and-play deployment and clear ablations for each MMU component.
- A classify-then-rewrite agent plus dual (continuous/discrete) generation yields parallel safe datasets, facilitating comparisons across architectures.
- On both HumanML3D and Motion-X, forget-set FID↑ and R@1↓ are stronger than LCR; gated α notably boosts retain-set R@1.
- Objectives (MPJPE/vel/acc/foot, alignment), random decoupling, and preservation divergence are spelled out, aiding reproducibility.

**Weaknesses:**

- “Forgetting” is evidenced mainly by larger FID or lower R@1 on unsafe splits. These can also reflect generic quality collapse (e.g., stationary outputs), which the paper itself observes, suggesting the proxy may over-credit safety. Stronger audits (human ratings, expert panels, adversarial prompts) are missing.
- The SafeMoEngine both labels and rewrites prompts and then generates replacement motions that populate the training/eval sets. Without human QC, leakage and confirmation bias are likely; the “safety” definition is primarily what the classifier rewrites detect. Release/readiness and license compatibility are not detailed.
- Foot-sliding/skating and low-amplitude/stationary behaviors are acknowledged, but quantitative physics proxies (contact precision, penetration, kinematic smoothness beyond Lfoot) or user studies are absent.
- SafeMo-Gated relies on an external text classifier at inference; misclassification risks either under- or over-suppression with limited analysis of false-positive/negative impacts. α-sensitivity and LoRA-rank sensitivity are only partially explored.
- Baselines focus on LCR and codebook manipulations; broader comparisons (e.g., alternate continuous-space editing, RL-based safety for T2M) are largely qualitative.

**Questions:**

1. Can you include blinded human safety/realism ratings and adversarial prompt suites to disentangle “distributional shift” from genuine safety? Any quantitative test where SafeMo outputs are explicitly non-violent yet semantically matched?
2. What is the accuracy/recall of your LLM classifier on an external, human-labeled safety set? How sensitive are results to false positives/negatives in the gating policy?
3. Beyond L_{\text{foot}}, can you report contact metrics, energy regularization, and motion smoothness post-negation (and across α)? Any success with post-filters or small finetunes to recover kinematics?
4. Will SafeMoVAE/VQ-29K include human-verified rewrites and motion validations? Please clarify licensing and provenance for all regenerated clips.
5. Results with other backbones (e.g., MLD) or LoRA ranks? α-sweep curves on retain/forget trade-offs? Does negation degrade OOD benign prompts (long paragraphs, figurative language)?

---

### Official Review · Reviewer_PLbK · 2025-10-31

**Soundness:** 3
**Presentation:** 3
**Contribution:** 2
**Rating:** 4
**Confidence:** 4

**Summary:**

This paper addresses the critical safety problem in text-to-motion (T2M) AIGC models, specifically the generation of harmful or violent motions. The authors argue that existing unlearning methods based on discrete tokens (e.g., VQ-VAE) are flawed, as they degrade the quality of benign motions and cause jerky transitions. To solve this, the paper proposes SafeMo, a framework with two main contributions: 1) SafeMoEngine, an LLM Agent-based pipeline to build the first safe T2M dataset (SafeMoVAE-29K); and 2) Minimal Motion Unlearning (MMU), a novel two-stage unlearning strategy that operates in continuous space, removing unsafe motions by "absorbing and negating" a harmful task vector. Experiments show that this method significantly outperforms the SOTA (LCR) in forgetting harmful motions while maintaining generation quality for safe motions.

**Strengths:**

- **Methodological Novelty**: MMU is a novel unlearning technique. By operating in continuous space, it avoids the inherent flaws of discrete methods. Its two-stage design is clever, particularly the use of a "negative preservation divergence" ($\mathcal{L}_{pres}$) in Stage 1 to deliberately degrade safe-task performance to isolate a pure harmful vector. This is technically sound.

- **Dataset Contribution**: The SafeMoEngine and the SafeMoVAE-29K dataset are valuable contributions to the T2M community. It provides the first standardized benchmark for the sub-field of "Safe T2M," and its "classify-then-rewrite" pipeline is more robust than simple keyword filtering.

- **Strong Quantitative Results**: The experimental results (especially in Tables 2 and 3) are impressive. SafeMo-Gated achieves a massive FID increase on the forget set, providing strong evidence of effective unlearning, while maintaining a high SOTA-level performance on the retain set.

**Weaknesses:**

- **Unfair Comparison**: This is the most critical flaw of the paper. The authors claim their MMU algorithm is superior to the LCR algorithm. However, the experiment compares [MMU + DiP (continuous model)] vs. [LCR + MoMask/BAMM (discrete models)]. This changes two variables at once: (1) the unlearning algorithm and (2) the model architecture. Therefore, it is impossible to know how much of the performance gain is due to the MMU algorithm versus the fact that DiP (2024) is simply a more advanced, continuous model that might be inherently more "editable" than older discrete models. This key confounding variable is not disentangled.

- **Missing Critical Baseline**: The paper effectively proposes two solutions for safety: (A) create a clean dataset with SafeMoEngine and train on it; and (B) unlearn from a "dirty" pre-trained model using MMU. The authors show the results for (A) on discrete models ("MoMask $D_r$"), but they critically omit the baseline for (A) on their own backbone: the "DiP $D_r$" (a DiP model trained from scratch on the authors' own clean SafeMoVAE-29K dataset). This is a major logical gap. If this simpler "DiP $D_r$" baseline performs as well as or better than the complex SafeMo-Gated, the necessity of the MMU method is undermined.

**Questions:**

- Can you provide a fairer comparison to prove the superiority of the MMU algorithm itself? For example, by implementing the LCR idea (or another unlearning method) on the continuous DiP model and comparing it to MMU on the same backbone.

- Why was the critical baseline "DiP $D_r$" (the DiP model trained from scratch on your new SafeMoVAE-29K clean dataset) omitted from the paper? How does its performance compare to SafeMo-Gated (which uses MMU)?

- As the authors, what is your recommended use case for your two different contributions (SafeMoEngine vs. MMU)? If a team has the resources to train a model from scratch, should they just use the SafeMoEngine-generated dataset, or should they train on the full dataset and then apply MMU?

---

### Note · Authors · 2025-11-29

I have read and agree with the venue's withdrawal policy on behalf of myself and my co-authors.